# Ubiquitin ligase RNF20 coordinates sequential adipose thermogenesis with brown and beige fat-specific substrates

Yong Geun Jeon [1], Hahn Nahmgoong[1], Jiyoung Oh[2], Dabin Lee[3], Dong Wook Kim[3], Jane Eunsoo Kim[1], Ye Young Kim [1], Yul Ji[1], Ji Seul Han [1], Sung Min Kim[1], Jee Hyung Sohn[1], Won Taek Lee [1], Sun Won Kim [1], Jeu Park[1], Jin Young Huh[1,4], Kyuri Jo [5], Je-Yoel Cho [3], Jiyoung Park [2] & Jae Bum Kim [1] ✉

In mammals, brown adipose tissue (BAT) and inguinal white adipose tissue (iWAT) execute sequential thermogenesis to maintain body temperature during cold stimuli. BAT rapidly generates heat through brown adipocyte activation, and further iWAT gradually stimulates beige fat cell differentiation upon prolonged cold challenges. However, fat depot-specific regulatory mechanisms for thermogenic activation of two fat depots are poorly understood. Here, we demonstrate that E3 ubiquitin ligase RNF20 orchestrates adipose thermogenesis with BAT- and iWAT-specific substrates. Upon cold stimuli, BAT RNF20 is rapidly downregulated, resulting in GABPα protein elevation by controlling protein stability, which stimulates thermogenic gene expression. Accordingly, BAT-specific Rnf20 suppression potentiates BAT thermogenic activity via GABPα upregulation. Moreover, upon prolonged cold stimuli, iWAT RNF20 is gradually upregulated to promote de novo beige adipogenesis. Mechanistically, iWAT RNF20 mediates NCoR1 protein degradation, rather than GABPα, to activate PPARγ. Together, current findings propose fat depot-specific regulatory mechanisms for temporal activation of adipose thermogenesis.

In mammals, adipose tissue plays a key role in regulating energy homeostasis and maintaining body temperature[1,2]. Adipose tissues are largely divided into white adipose tissue (WAT), which stores extra energy in the form of triglycerides, and brown adipose tissue (BAT), which produces heat through uncoupling protein 1 (UCP1) in the mitochondria[3]. In rodents, brown-like thermogenic adipocytes called beige (or brite) adipocytes are induced in inguinal WAT (iWAT) by hormonal and metabolic stimuli[4–6]. Catabolic capacity of thermogenic adipocytes has attracted considerable attention as a potential approach to counteract metabolic diseases[7–9].

In response to cold stimuli, mammals sequentially activate adipose thermogenesis for survival. At the acute phase of cold stimuli (~hours), BAT rapidly stimulates heat generation by activating brown adipocytes[3,10], thereby playing the first line of defence against hypothermia. Further, in the presence of chronic cold stimuli (>1 day), beige adipocytes are newly differentiated from a subset of preadipocytes

[1]Center for Adipocyte Structure and Function, Institute of Molecular Biology and Genetics, School of Biological Sciences, Seoul National University, Seoul 08826, South Korea. [2]Department of Biological Sciences, College of Information and Bioengineering, Ulsan National Institute of Science and Technology, Ulsan 44919, South Korea. [3]Department of Biochemistry, BK21 PLUS Program for Creative Veterinary Science Research and Research Institute for Veterinary Science, College of Veterinary Medicine, Seoul National University, Seoul 08826, South Korea. [4]Department of Life Science, Sogang University, Seoul 04107, South Korea. [5]Department of Computer Engineering, Chungbuk National University, Cheongju 28644, South Korea. ✉e-mail: jaebkim@snu.ac.kr

expressing PDGFRβ/CD81/BST2 in iWAT to increase wholebody thermogenic capacity[6,11,12], forming the second line of the thermoregulatory defence system. As each fat depot arises from distinct developmental origins[13] and reacts differently to cold stimuli, thermogenic execution of brown and beige fat might be differently regulated by fat depot-specific mechanisms. Although many recent studies have elucidated the origin and key factors of brown and beige fat thermogenesis[12,14–21], tissue-specific and temporal regulatory mechanisms of brown and beige fat thermogenesis upon cold exposure periods have not been well elucidated.

Emerging evidence suggests that E3 ubiquitin ligase regulates its substrates in a stimulus-dependent and tissue-specific manner[22–24]. Recently, it has been reported that E3 ubiquitin ligase ring finger protein 20 (RNF20) plays various roles in metabolic tissues. In WAT, RNF20 promotes nuclear corepressor 1 (NCoR1) degradation, thereby activating peroxisome proliferator-activated receptor gamma (PPARγ) and stimulating white adipocyte differentiation[25,26]. Also, in liver and kidney cancer, RNF20 promotes protein degradation of sterol regulatory element binding protein 1c (SREBP1c) to regulate lipid metabolism[27–29]. Further, RNF20 also promotes ubiquitination of other substrates such as histone H2B[30], Eg5[31], and eEF1BδL[32] in a cell type- and stimulus-dependent manner, raising the possibility that RNF20 might play distinct roles in brown and beige fat cells for adipose thermogenesis.

In this study, we demonstrate that RNF20 controls brown and beige fat thermogenesis with fat depot-specific substrates during cold stimuli. Upon acute cold stimuli, BAT RNF20 facilitates rapid thermogenic activation by guanine and adenine-binding protein alpha (GABPα) protein stability control. In addition, upon chronic cold stimuli, iWAT RNF20 potentiates beige adipocyte differentiation through NCoR1 degradation other than GABPα, which stimulates PPARγ activity for adipogenesis. Together, our data suggest that the sequential execution of adipose thermogenesis is exquisitely regulated by fat-depot-specific RNF20 substrates.

## Results

### Upon acute cold, BAT RNF20 downregulation potentiates thermogenic activity

Recently, it has been reported that RNF20 potentiates white adipocyte differentiation[25,26]. When we examined the potential roles of RNF20 in brown adipocyte differentiation, no significant changes were observed in lipid accumulation or expression of adipogenic markers by RNF20 suppression in brown preadipocytes (Supplementary Fig. 1a–d), implying that RNF20 might not have a profound effect on brown adipogenesis, which appears to be different from white adipogenesis[14,33]. Nonetheless, the mRNA level of *Ucp1* was elevated by RNF20 suppression in differentiated brown adipocytes (Supplementary Fig. 1b, d), implying that RNF20 might contribute to thermogenic gene expression in brown adipocytes.

To investigate the physiological roles of RNF20 in brown adipocytes, we examined the RNF20 level during cold stimuli. When differentiated brown adipocytes were treated with β-adrenergic agonists, such as isoproterenol, the level of RNF20 protein was rapidly downregulated, which preceded an increase in UCP1 (Fig. 1a). Furthermore, with cold stimuli, RNF20 protein was quickly decreased and its substrate NCoR1 protein was increased in BAT (Fig. 1b and Supplementary Fig. 2a, b). Conversely, when mice were exposed to warm environments after cold stimuli, the level of RNF20 was greatly upregulated in BAT (Supplementary Fig. 2c), suggesting that cold stimuli would decrease BAT RNF20. Also, we found that *Rnf20* was downregulated in high thermogenic brown adipocytes[34] compared to low thermogenic brown adipocytes (Supplementary Fig. 2d–f), proposing the possibility that RNF20 might be negatively related to the thermogenic activity of brown adipocytes. To address this, RNF20 expression was modulated in differentiated brown adipocytes. RNF20 overexpression (OE)

decreased the expression of thermogenic genes such as *Ucp1* and *Pgc1a* and attenuated oxygen consumption rate (OCR) in brown adipocytes (Fig. 1c–e). In contrast, *Rnf20* knockdown (KD) in differentiated brown adipocytes greatly stimulated thermogenic gene expression and oxygen consumption (Fig. 1f–h). These results suggest that RNF20 would suppress thermogenic activity in brown adipocytes.

Next, we investigated in vivo roles of RNF20 in BAT thermogenesis. As *Rnf20* wholebody knockout mice are embryonically lethal[35], we examined thermogenic activity in *Rnf20* defective (*Rnf20*[+/−]) mice. Under room temperature (RT) conditions, BAT of *Rnf20*[+/−] mice exhibited a relatively dark brown colour with abundant mitochondria, accompanied by small lipid droplets (LDs) compared to wild-type (WT) littermates (Fig. 1i and Supplementary Fig. 3a). Consistently, the expression levels of mitochondrial and thermogenic genes, as well as mitochondrial copy number, were enhanced in BAT of *Rnf20*[+/−] mice (Fig. 1j–l). Further, compared to WT mice, upon cold stimuli, *Rnf20*[+/−] mice exhibited higher rectal and BAT temperatures, accompanied by elevated wholebody heat generation and oxygen consumption (Fig. 1m–r). In addition, *Rnf20*[+/−] mice housed under thermoneutral (TN) conditions showed enhanced thermogenic capacity during cold stimuli (Supplementary Fig. 3b–d). Together, these data propose that cold-induced RNF20 downregulation would potentiate thermogenic activity in BAT.

### BAT-specific RNF20 modulation controls wholebody thermogenic activity upon acute cold stimuli

Since *Rnf20*[+/−] mice harbour RNF20 defects in all tissues, we attempted to modulate RNF20 expression in a BAT-specific manner. Injection of RNF20-expressing plasmid into BAT selectively increased the levels of RNF20 mRNA and protein in BAT, but not in other tissues (Fig. 2a–c). In BAT of RNF20 OE mice, the levels of thermogenic gene expression and mitochondrial copy number tended to decrease (Fig. 2d–f, and Supplementary Fig. 4a). In the presence of cold stimuli, BAT RNF20 OE mice exhibited decreased rectal and BAT temperatures (Fig. 2g–j). Consistent with these, heat generation and oxygen consumption were reduced in BAT RNF20 OE mice accompanied by 'whitening' of BAT with enlarged LDs (Fig. 2k, l, and Supplementary Fig. 4b–g), indicating that elevated RNF20 in BAT would suppress thermogenic execution.

To affirm that RNF20 would regulate BAT thermogenic activity, RNF20 was selectively knocked down in BAT by siRNA injection (Fig. 2m, n). In contrast to BAT RNF20 OE mice, BAT RNF20 KD potentiated thermogenesis upon cold stimuli (Fig. 2o–r). Moreover, RNF20 KD increased the expression of thermogenic and mitochondrial genes with small LDs (Supplementary Fig. 4h–k). These data suggest that BAT RNF20 downregulation would be important for conserving body temperature during acute cold stimuli.

### In BAT, RNF20 regulates thermogenesis via GABPα, a crucial factor for thermogenic activation

To elucidate the underlying mechanism(s) by which BAT RNF20 would control thermogenic activity, BAT proteomes of WT and *Rnf20*[+/−] mice were examined. Proteomics analysis showed that BAT-enriched mitochondrial proteins, including NDUFV3 and TOMM40, were upregulated in BAT of *Rnf20*[+/−] mice, whereas the levels of WAT-enriched genes regulating lipid storage were downregulated, accompanied by NCoR1 accumulation (Fig. 3a–c, Supplementary Fig. 5a–o), indicating that BAT RNF20 would preferentially regulate thermogenic activity.

As RNF20 is located in the nucleus and regulates the activity of several transcription factors[25,26,28,32,36], gene network analyses were performed[37,38] to identify key thermogenic target(s) in BAT of *Rnf20*[+/−] mice. Notably, we found that the upregulated genes in BAT of *Rnf20*[+/−] mice were mainly regulated by GABPα (Fig. 3d, e), a well-known transcription factor for mitochondrial biogenesis[39–41]. qRT-PCR validated that the pattern of mRNA expression of GABPα-regulated mitochondrial genes was enhanced in BAT of *Rnf20*[+/−] mice (Fig. 3f), implying

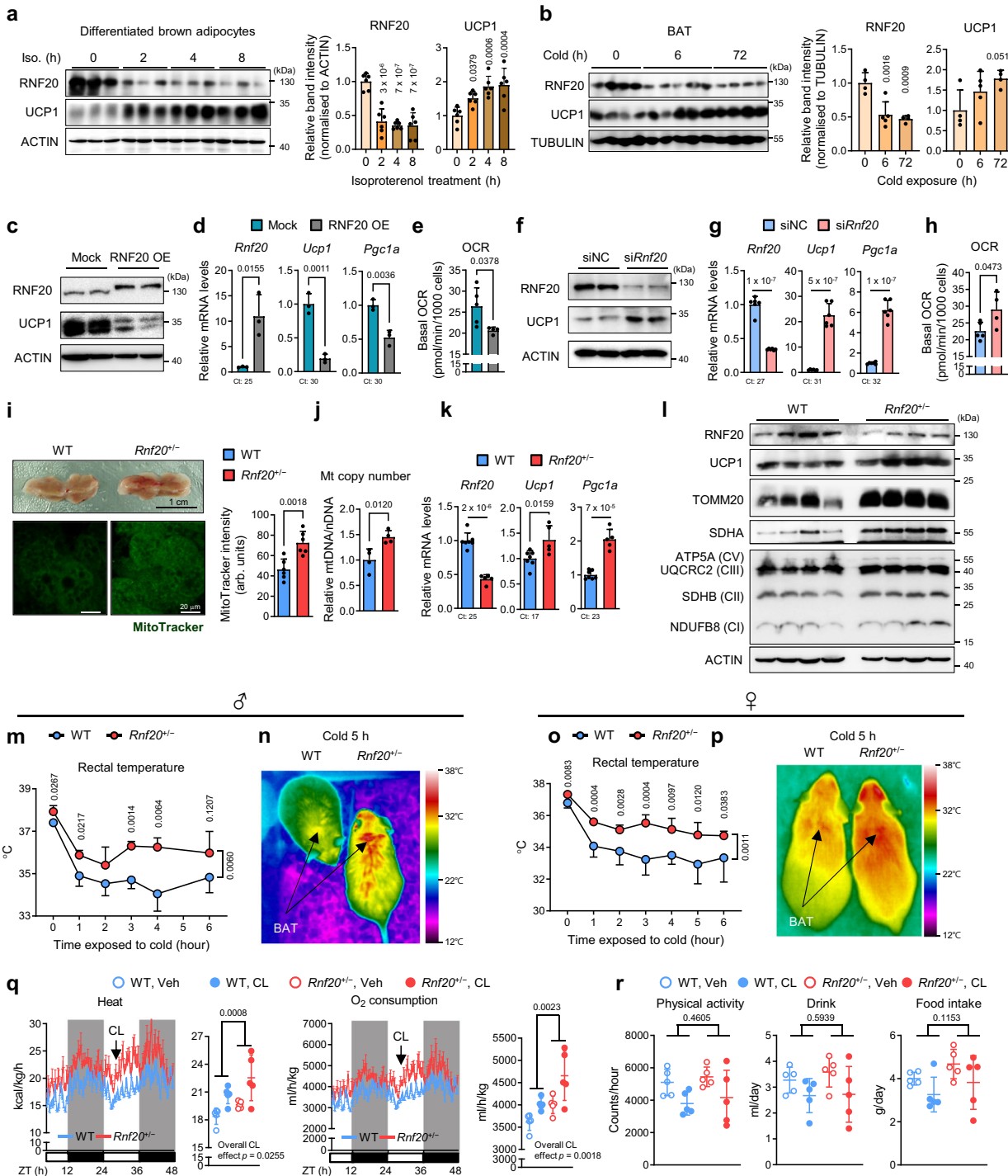

that RNF20 defects in BAT would facilitate transcriptional rewiring by stimulating GABPα activity.

As it has not been explored whether GABPα might regulate thermogenic activity in BAT, we decided to examine whether GABPα would potentiate BAT thermogenesis. Transcriptomic analysis revealed that genes upregulated by acute cold stimuli would be primarily modulated by GABPα in BAT (Fig. 3g, h, and Supplementary Fig. 6a). Moreover, single-cell RNA sequencing analysis showed that high thermogenic brown adipocyte-enriched genes seemed to be regulated by GABPα (Fig. 3i, j), proposing that BAT GABPα might upregulate thermogenic genes. Next, we explored whether GABPα might directly potentiate expression of thermogenic genes such as *Ucp1* and *Pgc1a*. Motif analysis showed that there would be a putative GABPα-binding motif in

the enhancer region of *Ucp1* and *Pgc1a* (Fig. 3k, Supplementary Fig. 6b, c), which was affirmed by luciferase assays and chromatin immunoprecipitation assays (Fig. 3l, Supplementary Fig. 6d, e). Accordingly, *Gabpa* suppression in differentiated brown adipocytes decreased the mRNA levels of *Ucp1, Pgc1a*, and *Dio2* (Supplementary Fig. 6f). In addition, in human brown adipocytes, *GABPA* expression was positively correlated with *UCP1* enhancer activity (Fig. 3m). Further, BAT-specific GABPα OE upregulated UCP1 protein level and thermogenic activity upon cold exposure (Fig. 3n, o). Thus, these data suggest that GABPα would be a crucial activator for BAT thermogenesis.

Next, we raised the question that elevated thermogenic activity in *Rnf20* defective mice might depend on GABPα stimulation. To address this, we suppressed GABPα in WT and *Rnf20* defective mice through

**Fig. 1 | Upon Acute Cold, BAT RNF20 Downregulation Potentiates Thermogenic Activity. a** Western blot analysis of a differentiated brown preadipocyte cell line (BAC) treated with isoproterenol (Iso. 1 μM) and relative band intensity normalised to ACTIN. The *p*-value was calculated compared to the 0 h group. *n* = 6 independent replicates. Representative results from two independent experiments. **b** Western blot analysis of brown adipose tissue (BAT) from mice housed at room temperature (RT, *n* = 4 mice) and exposed to a cold environment (6 °C, *n* = 5 (6 h), *n* = 4 mice (72 h)). Representative results from three independent experiments. The *p*-value was calculated compared to the 0 h group. **c, d** qRT-PCR and western blotting analyses of differentiated BAC cells transfected with mock or RNF20-expressing plasmid. *n* = 3 independent replicates. Ct: critical threshold. **e** Oxygen consumption rate (OCR) of differentiated brown adipocytes transfected with mock or RNF20-expressing plasmid. *n* = 5 (Mock), *n* = 4 mice (RNF20 OE). **f, g** qRT-PCR and western blot analyses of differentiated BAC cells transfected with siNC or si*Rnf20*. *n* = 3 independent replicates. **h** OCR of differentiated BAC cells transfected with siNC or si*Rnf20*. *n* = 5 independent replicates (siNC), *n* = 4 independent replicates (si*Rnf20*). **i** Representative macroscopic view, wholemount MitoTracker staining images, and quantitative analysis of MitoTracker intensity of BAT of WT and *Rnf20*[+/−] mice housed under RT. **j–l** Western blot analysis, qRT-PCR analysis (*n* = 4 (WT), *n* = 4 (*Rnf20*[+/−]) mice), and mitochondrial DNA (mtDNA) (*n* = 7 (WT), *n* = 5 (*Rnf20*[+/−]) mice) content of genes related to thermogenesis in BAT of WT and *Rnf20*[+/−] mice housed under RT. Representative results from two independent experiments. **m–p** Rectal temperature and representative infrared images of male (*n* = 4 (WT), *n* = 4 (*Rnf20*[+/−]) mice) and female (*n* = 8 (WT), *n* = 4 (*Rnf20*[+/−]) mice) WT and *Rnf20*[+/−] mice upon cold exposure (6 °C). For (**m, n**) representative results from three independent experiments. **q, r** Wholebody heat generation, oxygen consumption, physical activity, drink, and food intake in male WT and *Rnf20*[+/−] mice housed under RT. During the experiment, β3-adrenergic agonist CL316,243 (CL, 0.5 mg/kg body weight) was administrated (arrow). *n* = 5 mice. Source data are provided as a Source Data file. *n* indicates biological replicates. Data are represented as mean ± S.D. Significance was determined using one-way ANOVA with Dunnett's multiple comparison (**a, b, q**), unpaired two-sided Student's *t*-test (**d, e, g, h, i, j, k**), repeated measures ANOVA with Tukey's multiple comparisons test (**m, o, q**), and two-way ANOVA (**q, r**).

siRNA injection. GABPα KD in BAT nullified the increased UCP1 protein and the enhanced thermogenic activity of *Rnf20* defective mice upon cold stimuli (Fig. 3p, q and Supplementary Fig. 6g). Interestingly, we found that the level of GABPα protein was elevated in BAT of *Rnf20* defective mice (Fig. 3p), implying that RNF20 might regulate GABPα protein stability. Together, these data suggest that BAT RNF20 would control acute thermogenic activity by regulating GABPα.

### RNF20 promotes GABPα protein degradation in BAT
E3 ubiquitin ligase RNF20 promotes degradation of its substrates[25,29,36,42], which prompted us to explore whether BAT RNF20 might affect GABPα degradation. As shown in Fig. 4a, GABPα protein was highly expressed in BAT compared to that in iWAT. In addition, the level of GABPα protein, but not its mRNA, was upregulated by cold stimuli in BAT (Fig. 4b, c), which was negatively correlated with RNF20. Moreover, RNF20 OE in BAT reduced the level of GABPα protein, whereas RNF20 suppression increased GABPα protein (Fig. 4d, f, h). Nonetheless, mRNA levels of *Gabpa* were not altered by RNF20 modulation (Fig. 4e, g, i), leading us to speculate that RNF20 would mediate GABPα protein degradation.

As endogenous RNF20 bound to endogenous GABPα in BAT (Fig. 4j), we decided to test whether RNF20 might stimulate GABPα protein degradation. As indicated in Fig. 4k, l, RNF20 OE facilitated polyubiquitination of GABPα in BAT, whereas RNF20 deficiency decreased the degree of GABPα polyubiquitination. Furthermore, the proteasome inhibitor MG132 relieved GABPα degradation by RNF20 (Fig. 4m), and cycloheximide chasing experiments in differentiated brown adipocytes showed that RNF20 enhanced GABPα degradation (Fig. 4n, o). Thus, these data propose that RNF20 would stimulate GABPα degradation in BAT.

Since GABPα is a transcription factor[40], we then explored whether RNF20 would affect GABPα activity. RNF20 OE repressed GABPα activity at its canonical target genes such as *Mtif2*[43] and *Ucp1*, whereas RNF20 KD potentiated GABPα activity (Fig. 4p–s), implying that BAT RNF20 could regulate thermogenic gene expression by facilitating proteasomal degradation of GABPα.

### In iWAT, RNF20 potentiates beige adipocyte thermogenesis upon chronic cold stimuli, independent of GABPα
Upon chronic cold exposure (e.g., >1 day), the differentiation of beige adipocytes, characterised by multilocular LDs, was boosted in iWAT to sustain wholebody thermogenesis[44] (Fig. 5a). To explore the role of RNF20 in beige fat thermogenesis, we examined RNF20 levels during cold exposure. Unexpectedly, unlike BAT, the level of RNF20 protein in iWAT was upregulated upon chronic cold stimuli (Fig. 5b and Supplementary Fig. 7a), with a little alteration of GABPα, implying that there might be iWAT-specific regulation of RNF20 with its substrates upon cold stimuli.

To address this, RNF20 was specifically overexpressed in iWAT in a contralateral manner and then exposed to chronic cold (Fig. 5c, d). When iWAT temperature was monitored with a thermal camera, the temperature of iWAT area with RNF20 OE appeared to be higher than that in the mock control (Fig. 5e and Supplementary Fig. 7b, c). Moreover, RNF20 OE in iWAT increased the formation of multilocular adipocytes and potentiated the expression of thermogenic and fatty acid oxidation genes with slightly enhanced insulin sensitivity upon chronic cold stimuli (Fig. 5f–h and Supplementary Fig. 7d–m), suggesting that RNF20 would stimulate beige fat thermogenesis upon chronic cold challenges. Conversely, iWAT of *Rnf20* KD and *Rnf20* defective mice showed fewer multilocular adipocytes than control groups, and expression levels of thermogenic genes were downregulated upon chronic cold stimuli (Fig. 5i–k and Supplementary Fig. 8a–f). Intriguingly, iWAT RNF20 OE did not largely alter GABP protein levels without largely affecting GABPα polyubiquitination or protein degradation rate (Fig. 5l–n and Supplementary Fig. 8g–l). Together, these data propose that RNF20 in iWAT would facilitate beige fat thermogenesis upon chronic cold stimuli, probably, by utilizing different substrates other than GABPα.

### iWAT RNF20 potentiates beige fat thermogenesis by promoting NCoR1 degradation to stimulate PPARγ
Given that RNF20 potentiates PPARγ activity in WAT by promoting NCoR1 protein degradation[25,26], we decided to test the possibility that RNF20 would stimulate beige fat cell formation by regulating NCoR1-PPARγ axis. In iWAT, the mRNA levels of PPARγ target genes were strongly induced by chronic cold stimuli accompanied by NCoR1 downregulation (Fig. 6a, b, and Supplementary Fig. 9a, b). Since the function of NCoR1 in beige fat thermogenesis is not fully understood, we overexpressed NCoR1 specifically in iWAT (Fig. 6c). iWAT-selective NCoR1 OE repressed thermogenic activity, accompanied by downregulation of PPARγ target genes, including *Ucp1* (Fig. 6c–e and Supplementary Fig. 9c). Conversely, iWAT-selective NCoR1 KD enhanced thermogenic activity in iWAT (Supplementary Fig. 9d), indicating that NCoR1 in iWAT would suppress beige fat thermogenesis by inhibiting PPARγ.

To test the idea that RNF20 would facilitate beige fat thermogenesis through NCoR1 protein regulation, we examined NCoR1 protein level upon RNF20 OE. Upon chronic cold, RNF20 OE in iWAT decreased NCoR1 protein level (Fig. 6f) with increased NCoR1 polyubiquitination and facilitated NCoR1 protein degradation in beige adipocytes (Supplementary Fig. 9e, f), which in turn stimulated PPARγ target gene expression (Fig. 6f, g and Supplementary Fig. 9g–j),

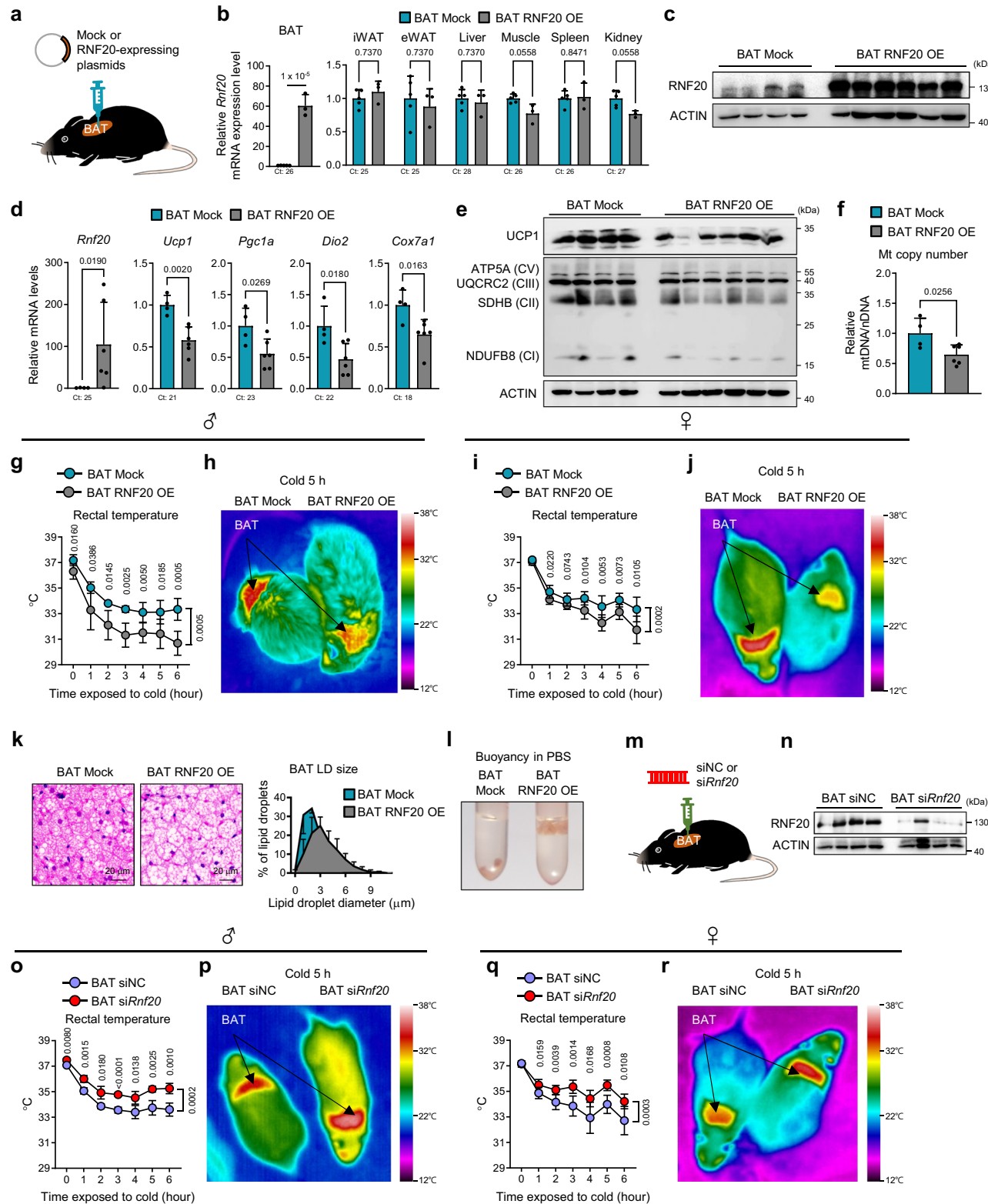

implying that iWAT RNF20 would determine the degree of NCoR1 protein and PPARγ activity upon chronic cold stimuli. In accordance with these, RNF20 and NCoR1 double OE showed that RNF20 OE partly alleviated the inhibitory effect of NCoR1 OE on beige fat thermogenesis (Fig. 6h–j and Supplementary Fig. 9k). In line with these, *Rnf20* and *Ncor1* double KD partly nullified the effect of *Rnf20* on iWAT thermogenic activity (Supplementary Fig. 9l). Together, these data suggest

that upon chronic cold stimuli, iWAT RNF20 would promote NCoR1 degradation to stimulate PPARγ for beige fat thermogenesis.

### In iWAT, RNF20 stimulates de novo beige adipogenesis upon chronic cold stimuli

Upon chronic cold stimuli, beige adipocytes predominantly arise from beige progenitors[44,45]. Consistent with these, chronic cold stimuli

**Fig. 2 | BAT-specific RNF20 Modulation Controls Thermogenic Activity Upon Acute Cold Stimuli. a** Experimental scheme for plasmid injection (5 μg) into BAT with in vivo jetPEI. **b** qRT-PCR of BAT and other peripheral tissues in mock and RNF20 overexpression (OE) mice. $n = 5$ (Mock), $n = 3$ (RNF20 OE) mice. **c–f** Western blotting, qRT-PCR, mtDNA content of BAT from mock and RNF20 OE mice housed under RT. $n = 4$ (Mock), $n = 6$ (RNF20 OE) mice. Representative results from two independent experiments. **g–j** Rectal temperature and representative infrared images of mock and RNF20 OE male ($n = 6$ (Mock), $n = 6$ (RNF20 OE) mice) and female mice ($n = 8$ (Mock), $n = 7$ (RNF20 OE) mice) upon cold exposure (6 °C). Representative results from three independent experiments. **k** Representative H&E images of BAT from mock and RNF20 OE mice and the distribution of the LD size of BAT. **l** Representative image showing BAT of RNF20 OE mice floating in phosphate-buffered saline (PBS; density: 1.0723 g/cm³). **m** Experimental scheme for siRNA injection (5 μg) into BAT. **n** Western blot analysis of BAT of siNC or si*Rnf20* mice housed under RT. Representative results from two independent experiments. **o–r** Rectal temperature and representative infrared images of siNC or si*Rnf20* male ($n = 5$ (siNC), $n = 4$ (si*Rnf20*) mice) and female mice ($n = 7$ (siNC), $n = 8$ (si*Rnf20*) mice) upon cold exposure (6 °C). In vivo experiments were performed 3 d after nucleotide injection. Representative results from two independent experiments. Source data are provided as a Source Data file. $n$ indicates biological replicates. Data are represented as mean ± S.D. Significance was determined using unpaired two-sided Student's $t$-test (**b, d, f**), multiple unpaired $t$-test with False Discovery Rate (**b**), and repeated measures ANOVA with Tukey's multiple comparisons test (**g, i, o, q**).

induced the expression of adipogenic genes (Fig. 7a). The fact that RNF20 promotes white adipocyte differentiation by regulating NCoR1-PPARγ axis[25] led us to speculate that RNF20 would stimulate beige adipocyte differentiation. When platelet-derived growth factor receptor alpha-expressing (CD31⁻CD45⁻PDGFRα⁺) preadipocytes[46] were induced to differentiate into beige adipocytes, RNF20 protein levels were upregulated (Fig. 7b and Supplementary Fig. 10a, b). To investigate whether RNF20 might promote beige fat cell differentiation, PDGFRα⁺ preadipocytes were overexpressed with RNF20 and differentiated into beige adipocytes (Fig. 7c). RNF20 OE facilitated LD formation and the expression of thermogenic genes such as *Ucp1* and *Pgc1a* (Fig. 7d, e). Conversely, *Rnf20* suppression in PDGFRα⁺ preadipocytes repressed beige fat cell differentiation (Fig. 7f–h and Supplementary Fig. 10c–e). In line with these, RNF20 OE in mature beige adipocytes augmented expression levels of thermogenic genes whereas *Rnf20* KD decreased them (Supplementary Fig. 10f–i). Together, these data imply that iWAT RNF20 would promote beige fat cell differentiation upon stimuli.

To affirm that RNF20 might facilitate beige adipogenesis in vivo, we took advantage of AdipoChaser mouse model, in which newly differentiated adipocytes can be labelled by doxycycline administration[44,47] (Fig. 7i, j). RNF20 OE in AdipoChaser iWAT increased the number of newly differentiated beige adipocytes and the level of UCP1 protein upon chronic cold stimuli (Fig. 7k and Supplementary Fig. 10j). In contrast, RNF20 KD suppressed de novo beige adipogenesis upon chronic cold stimuli (Supplementary Fig. 10k, l). Together, these findings propose that upon prolonged cold stimuli, iWAT RNF20 would potentiate beige adipogenesis in vivo.

## Discussion

During cold stimuli, mammals execute spatiotemporally coordinated adipose thermogenesis. Upon acute cold stimuli, brown adipocyte thermogenesis is rapidly activated to prevent hypothermia[4,13]. On the other hand, in the presence of prolonged cold, shivering thermogenesis by the muscle is substantially reduced[48] and beige adipocyte formation is boosted to maintain body temperature[4,13]. Given that the origins and roles of two fat depots such as BAT and iWAT are distinct, it is feasible to speculate that adipose thermogenesis upon cold stimuli duration might be regulated by fat depot-specific mechanisms. However, since it is technically difficult to modulate genes in a fat depot-specific manner, in vivo studies to investigate distinct mechanisms of brown and beige fat have not been well explored. Here, using fat depot-specific genetic modulation, we demonstrated that E3 ubiquitin ligase RNF20 would control brown fat cell activation and beige fat cell differentiation through fat depot-specific substrates during cold stimuli. Our findings suggest that RNF20-GABPα axis in BAT would rapidly activate thermogenic activity upon acute cold stress, whereas upon chronic cold stimuli, RNF20-NCoR1-PPARγ axis in iWAT would protect against hypothermia by stimulating beige adipocyte differentiation. Thus, RNF20 would play a crucial role in orchestrating the timely activation of adipose thermogenesis upon cold stimuli (Fig. 8).

To prevent hypothermia, BAT needs immediately to activate thermogenesis, which is primarily achieved by increasing thermogenic gene expression and mitochondrial activity. It is well established that PKA and MAPK signalling pathways phosphorylate ATF2 and PGC1α to stimulate these transcriptional activators[3]. However, as phosphorylation is often temporal and transient, other relatively stable and rapid mechanisms would be required to potentiate mitochondrial biogenesis and thermogenic gene expression upon acute cold stimuli. Here, current data propose that E3 ubiquitin ligase RNF20 in BAT contributes to acute thermogenic activation by repressing ubiquitin-mediated protein degradation of GABPα. Upon acute cold stimuli, RNF20 was suppressed to potentiate BAT thermogenic activity, whereas, under warm conditions, BAT RNF20 promoted proteasomal degradation of GABPα. Furthermore, we demonstrated that GABPα stimulated expression of thermogenic genes and mitochondrial biogenesis in BAT. Consistently, RNF20 downregulation increased GABPα protein levels, which promoted transcription of mitochondrial and thermogenic genes to stimulate BAT thermogenesis. Moreover, in human brown adipocytes, *GABPA* expression levels are positively correlated with human *UCP1* enhancer activity. In addition, we observed that RNF20 could regulate NCoR1 protein abundance in BAT, which appeared to regulate lipid metabolism (Supplementary Fig. 5). Therefore, our data propose that RNF20 would be a key player for rapidly and reliably controlling BAT thermogenic activation by modulating GABPα protein.

Upon prolonged cold stimuli, beige adipocytes are differentiated in iWAT to facilitate thermogenic activity for survival[6,45]. It has been suggested that beige adipocytes in rodents primarily arise from beige progenitors upon cold exposure (i.e., de novo beige adipogenesis)[44,45]. Although PPARγ is a crucial factor for de novo beige adipogenesis, underlying mechanisms that stimulate PPARγ activity in response to prolonged cold stimuli are largely unknown. In this study, we demonstrated that upon chronic cold stimuli, an increase in RNF20 activated PPARγ by promoting NCoR1 degradation for beige adipogenesis. Several lines of evidence support this. First, iWAT-specific RNF20 OE potentiated thermogenic gene expression with abundant beige adipocytes. Second, iWAT-specific NCoR1 OE suppressed PPARγ target gene expression and beige fat thermogenesis. Third, RNF20 decreased NCoR1 protein levels, and thus, RNF20 OE in iWAT relieved the inhibitory effect of NCoR1 on beige fat thermogenesis. Lastly, in vitro and in vivo adipogenic experiments showed that RNF20 stimulated de novo beige adipogenesis in iWAT. Nevertheless, as we analysed systemic *Rnf20* heterozygous-null mice and fat depot-specific *Rnf20* modulation mice, it is plausible to speculate that RNF20 modulation in other cell types would affect thermogenic phenotypes. Thus, it is important to scrutinize brown adipocyte- or beige precursor-specific *Rnf20* OE and KO animal models to elucidate the role of RNF20 in brown adipocyte activation and beige adipocyte differentiation in future studies.

One of the intriguing observations in this study is that GABPα would be the primary target of RNF20 in BAT, but not in iWAT, which

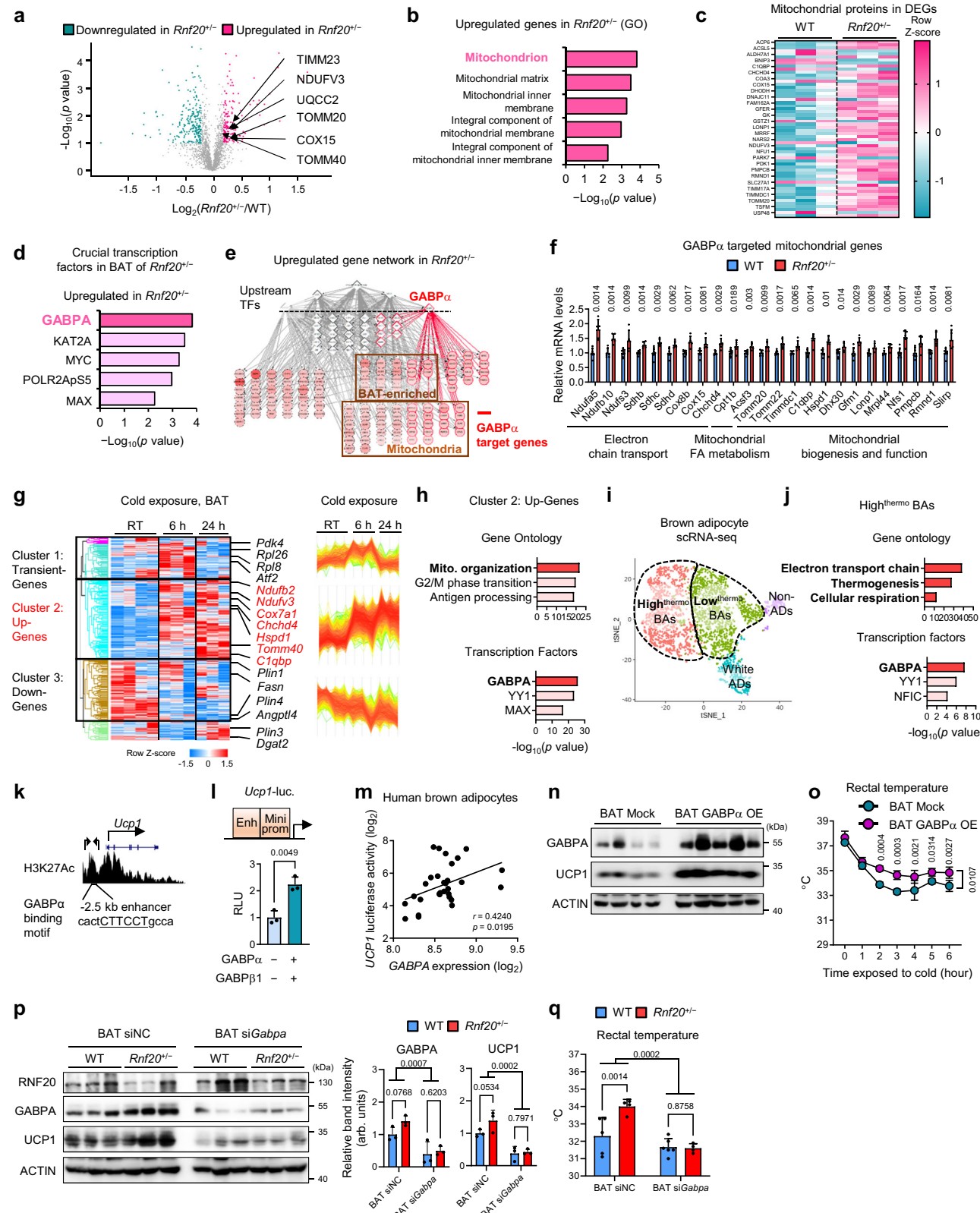

might probably enable RNF20 to regulate fat depot-selective thermogenic execution during cold exposure. It has been reported that several E3 ubiquitin ligases exhibit tissue-specific substrate preference[23,24,49–53]. For example, E3 ubiquitin ligase FBXW7 inhibits cell proliferation in several cancer cells by ubiquitylating MYC, whereas it promotes cell proliferation in certain tumours by p100 degradation[50].

Although underlying mechanisms have not yet been well elucidated, it is feasible that the abundance of tissue-specific substrates of E3 ubiquitin ligases might be associated with these phenomena. Similarly, MKRN1 promotes AMPKα1 protein degradation in the liver and adipose tissue[52], in which AMPKα1 is abundantly expressed[54,55]. However, MKRN1 does not appear to determine AMPKα1 abundance in the brain,

**Fig. 3 | In BAT, RNF20 Regulates Thermogenesis via GABPα. a–c** Volcano plot, gene ontology, and heatmap of the proteome from WT and *Rnf20*[+/−] mice. *n* = 3 mice. **d** Crucial transcription factors (TFs) of upregulated proteins. **e** Gene network analysis of upregulated proteins in BAT of WT and *Rnf20*[+/−] mice. GABPα target genes are labelled as red connection lines. **f** qRT-PCR analysis of GABPα-targeted mitochondrial genes in the BAT of WT and *Rnf20*[+/−] mice housed at RT. *n* = 7 (WT), *n* = 5 (*Rnf20*[+/−]) mice. **g** Gene clustering analysis of the BAT transcriptome (GSE119452). **h** Gene ontology and crucial transcription factors of the upregulated genes (cluster 2). **i** t-distributed stochastic neighbour embedding (tSNE) plot of BAT single-cell RNA-seq data (GSE125269). Adipocytes were largely divided into high[thermo] BAs, low[thermo] BAs, and white adipocytes (WAs). **j** Gene ontology and crucial transcription factors of high[thermo] BA-enriched genes. **k** Mouse *Ucp1* enhancer regions with H3K27Ac enrichment peaks in BAT (GSE63964). Primers used for enhancer cloning are indicated. **l** Luciferase activity of *Ucp1*-luciferase (luc.) constructs containing their enhancer (−2.5 kb upstream, Materials and Methods), minimal promoter (Mini prom), and TATA-box element. *n* = 3 independent replicates. **m** Correlation of *GABPA* expression level with *UCP1*-enhancer luciferase activity in differentiated clonal brown preadipocytes from human BAT (GSE68544). **n, o** Western blotting and rectal temperatures during cold exposure (6 °C) of mock and GABPA-OE mice housed under RT. *n* = 3 mice. Representative results from two independent experiments. **p** Western blotting analysis of BAT of WT, *Rnf20*[+/−], WT with si*Gabpa*, and *Rnf20*[+/−] with si*Gabpa* mice housed under RT. **q** Rectal temperatures of WT (*n* = 5), *Rnf20*[+/−] (*n* = 4), WT with si*Gabpa* (*n* = 6), and *Rnf20*[+/−] with si*Gabpa* (*n* = 4) mice during cold exposure (6 °C). In vivo experiments were performed 3 d after nucleotide injection. Source data are provided as a Source Data file. *n* indicates biological replicates. Data are represented as mean ± S.D. Significance was determined using multiple unpaired *t*-tests with False Discovery Rate (**a, f**), Fisher's exact test with Benjamini–Hochberg method (**b, d, h, j**), unpaired two-sided Student's *t*-test (**l**), two-tailed Pearson correlation (**m**), repeated measures ANOVA with Tukey's multiple comparisons test (**o**), and two-way ANOVA (**p, q**).

where AMPKα1 is expressed at low levels[52,54,55]. Our data showed that the level of GABPα protein was high in BAT, but relatively low in iWAT. Intriguingly, when GABPα was overexpressed in iWAT, the degree of polyubiquitination of GABPα was enhanced (Supplementary Fig. 11a, b), implying that the level of substrate might be one of the crucial factors in determining substrate specificity, at least partly, in RNF20 E3 ubiquitin ligase. On the other hand, it seems that RNF20 would target NCoR1 in both iWAT and BAT, speculating that substrate quantity might be involved in the target preference of E3 ubiquitin ligase. Nevertheless, as signalling cascades and substrate recognition receptors are able to affect the substrate specificity of E3 ubiquitin ligase[22,56–58], we cannot exclude the possibility that the abundance of substrates would not be the sole factor for determining tissue-specific protein degradation, which needs to be investigated in future. In addition, it needs to elucidate the mechanisms by which the levels of RNF20 proteins are differentially regulated upon cold duration in BAT and iWAT. In this regard, it has been reported that phosphoproteome upon cold stimuli is significantly different in brown and beige adipocytes[59]. In future, it will be necessary to investigate underlying mechanisms by which fat depot-specific signalling could regulate the level of RNF20 protein.

This study shows one of the key regulatory mechanisms by which adipose thermogenesis is spatiotemporally coordinated upon cold duration. Current data show that RNF20 differentially controls brown adipocyte activation and beige adipocyte differentiation via fat depot-preferential proteolysis of GABPα and NCoR1. Since each fat depot is likely to execute a unique mechanism that plays its own physiological roles in maintaining energy homeostasis, the fat depot-specific molecular mechanisms proposed in this study would provide an important perspective on adipose biology.

## Methods
### Animal experiments
In *Rnf20* defective (*Rnf20*[+/−]) mice, exons 3–20 of the *Rnf20* gene were deleted. *Rnf20*[+/−] mice were obtained from the knockout mouse project repository (KOMP). This mouse strain, [C57BL/6N-*Rnf20*[tm1(KOMP)Vlcg]/TcpMmucd (RRID:MMRRC_049486-UCD)], was obtained from the Mutant Mouse Resource and Research Centre at the University of California at Davis[35]. Mice were backcrossed with C57BL/6 J for more than 10 generations[25]. Mice were fed normal chow diet (13.12% of energy from fat, PicoLab Rodent Diet 20, LabDiet 5053, Texas), and housed in a temperature- and humidity-controlled (50%), specific pathogen-free animal facility at 22 °C, under a 12:12 h light:dark cycle, and health status checks were performed two or three times a week. Mice were mated in-house with WT and *Rnf20*[+/−] mice, and WT littermates and *Rnf20*[+/−] mice were used for genotyping (Supplementary Data 1). For the cold tolerance test, 3–4-month-old male and female mice were placed in a temperature-controlled rodent incubator maintained at 6 °C (Environmental Cabinet, DBL Co.). For thermoneutral and cold-exposure experiments, 3–4-month-old male mice were kept at 30 °C for 7 d, and then these mice were exposed to cold environments (6 °C) in a climate-controlled rodent incubator. Rectal temperature was measured using a thermal probe (Testo925, Testo Inc.). BAT Mock, RNF20 OE, and GABPA OE experiments were conducted on the same day (Figs. 2g, o and 3o), and BAT siNC and si*Rnf20* experiments (Fig. 2i, q) were conducted on the same day. An infrared camera was used to measure the body surface temperature of the mice (CX320 Thermal Imaging Camera; COX Co.). Mice were euthanized by carbon dioxide asphyxiation (CO2) inhalation.

AdipoChaser mice (Adipoq-rtTA; TRE-Cre; Rosa26-loxp-stop-loxp-YFP) were kindly provided by Dr. Philipp Scherer (UT Southwestern). For adipocyte chasing experiments[60], 12-week-old male AdipoChaser mice were fed a doxycycline (600 mg/kg)-containing chow diet for 2 weeks to label the old adipocytes, and then fed a normal chow diet.

For BAT- or iWAT-specific plasmid or siRNA delivery, 5 μg of plasmid or siRNA was directly injected into BAT or iWAT of male and female mice using In vivo-jetPEI® (201-10 G, Polyplus). Cold exposure experiments were performed 3 d after nucleotide injection. This study was reviewed and approved by the Institutional Animal Care and Use Committee of Seoul National University.

### Adipose tissue fractionation
BAT and iWAT were minced into small pieces and digested in collagenase buffer (0.1 M HEPES, 0.125 M NaCl, 5 mM KCl, 1.3 mM CaCl₂, 5 mM glucose, 1.5% (w/v) bovine serum albumin, and 0.1% (w/v) collagenase I) at 37 °C for 1 h with shaking.

### Fluorescence-activated cell sorting (FACS)
Stromal vascular fraction (SVFs) were stained with anti-CD31 (1:200, 102405, BioLegend), anti-CD45 (1:200, 103107, BioLegend), and anti-PDGFRα (1:100, 135902, BioLegend). The cells were analysed using a FACS Canto II instrument (BD Biosciences).

### Cell culture and differentiation
All cells were cultured at 37 °C and 5% CO₂ in a humidified cell incubator. To differentiate SVF-derived brown adipocytes, SVFs from BAT were grown to confluence in Dulbecco's modified Eagle's medium (DMEM) containing 10% FBS. After achieving confluent growth, the cells were induced with dexamethasone (1 μM), methylisobutylxanthine (520 μM), insulin (20 nM), indomethacin (125 μM), T3 (1 nM), and rosiglitazone (1 μM). After 2 d, the culture medium was replaced with DMEM containing 10% FBS, insulin (20 nM), T3 (1 nM), and rosiglitazone (1 μM), and the cells were incubated for 5–6 d. To differentiate iWAT PDGFRα-expressing (CD31⁻CD45⁻PDGFRα⁺) preadipocytes-derived beige adipocytes[6], PDGFRα-expressing

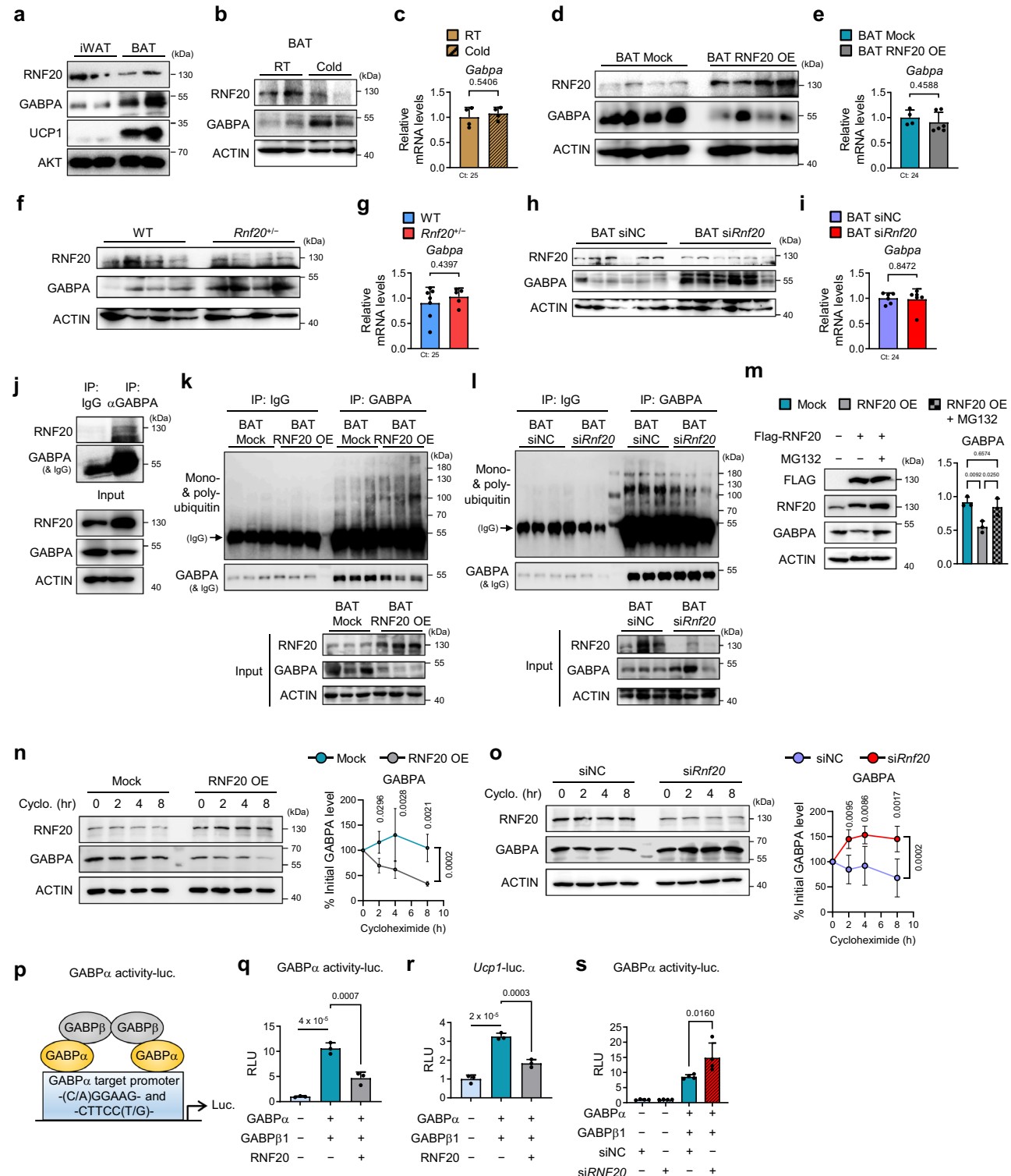

(CD31⁻CD45⁻PDGFRα⁺) preadipocytes from iWAT were grown to confluence in DMEM supplemented with 10% FBS. After achieving confluent growth, the cells were stimulated with dexamethasone (1 μM), methylisobutylxanthine (520 μM), insulin (850 nM), indomethacin (125 μM), T3 (1 nM), and rosiglitazone (1 μM). After 2 days, the culture medium was replaced with DMEM containing 10% FBS, insulin (850 nM), T3 (1 nM), and with or without rosiglitazone (1 μM) (described in the figure legend), and the cells were incubated for 5–6 d. To differentiate the immortalised brown preadipocyte cell line (BAC), which was provided by Dr. Kai Ge (National Institutes of Health)[20], cells

were stimulated with dexamethasone (1 μM), methylisobutylxanthine (520 μM), insulin (20 nM), indomethacin (125 μM), and T3 (1 nM). After 2 days, the culture medium was replaced with DMEM containing 10% FBS, insulin (20 nM), and T3 (1 nM), and the cells were incubated for 5–6 days.

## Isolation and separation of two brown adipocyte subpopulations

Brown adipocytes from interscapular BAT were obtained from 3–4-month-old male mice[34]. Briefly, BATs were minced into small pieces

**Fig. 4 | RNF20 Promotes GABPα Protein Degradation in BAT, A Crucial Factor for Thermogenesis. a** Western blotting analysis of inguinal WAT (iWAT) and BAT. **b, c** Protein and mRNA levels (n = 4 (RT), n = 4 (Cold) mice) of *Gabpa* in BAT from mice upon cold exposure (6 °C) for 6 h. **d, e** Protein and mRNA levels (n = 4 (Mock), n = 6 (RNF20 OE) mice) of *Gabpa* in BAT of mock and RNF20 OE mice. **f, g** Protein and mRNA levels (n = 7 (WT), n = 5 (*Rnf20*+/−) mice) of *Gabpa* in BAT of WT and *Rnf20*+/− mice. **h, i** Protein and mRNA levels of *Gabpa* (n = 6 (siNC), n = 6 (si*Rnf20*) mice) in BAT of mice injected with siNC or si*Rnf20*. For **a–i** representative results from two independent experiments. **j** Endogenous co-immunoprecipitation using GABPα antibody in BAT. **k** Ubiquitin levels of GABPα in BAT of mock and RNF20 OE. **l** Ubiquitin levels of GABPα in BAT of siNC and si*Rnf20*. Representative results from two independent experiments. **m** Protein level of GABPα transfected with RNF20-

expressing plasmids in HEK293T cells without or with MG132 treatment (20 μM for 6 h). **n, o** Cycloheximide (30 μM)-chasing assay of GABPα protein in differentiated brown adipocytes. n = 3 independent replicates. Representative results from three independent experiments. **o** Scheme for luciferase assay using mouse *Mtif2* promoter (−86 bp upstream) in which GABPα-binding motifs (C/AGGAAG or CTTCCT/G) are present (GABPα activity luc.). **p** GABPα activity luc. with GABPα and GABPβ1-expressing plasmids. n = 3 independent replicates. **q–s** Effects of RNF20 over-expression and knockdown on GABPα or *Ucp1*-luciferase activity. n = 3. Source data are provided as a Source Data file. *n* indicates biological replicates. Data are represented as mean ± S.D. Significance was determined using an unpaired two-sided Student's *t*-test (**c, e, g, i**), and one-way ANOVA with Tukey's multiple comparison (**m–o, q–s**).

and digested in collagenase buffer. The cell suspension was filtered through a 100 μm nylon filter (93100, SPL Life Sciences) to remove undigested tissues and resuspended in three volumes of DMEM containing 10% FBS, following which the filtrate was centrifuged at 200 × *g* for 8 min. Floating brown adipocytes were transferred and re-suspended in DMEM containing 10% FBS three times under the same centrifugation conditions to eliminate other SVF contaminants. The adipocyte suspension was then centrifuged at 100 × *g* for 5 min. The supernatant was carefully collected as a low-thermogenic brown adi-pocyte subpopulation. The rest of the cell mixture was resuspended in DMEM containing 10% FBS and centrifuged at 100 g for 5 min. After centrifugation, the thick top layer was the high thermogenic-brown adipocyte subpopulation.

### Western blot, immunoprecipitation, and quantification
Cells and tissues were lysed on ice with radioimmunoprecipitation assay (RIPA) buffer [150 mM NaCl, 50 mM Tris-HCl (pH 7.4), 1% NP-40, 0.25% sodium deoxycholate, 1 mM EDTA, protease inhibitor cocktail (P3100; GenDEPOT), and MG132 (BML-PI102; Enzo Life Sciences)]. Total cell lysates were obtained by centrifugation at 13,000 × *g* for 15 min at 4 °C, and 1 mg of lysates was used for immunoprecipitation. The lysates were pre-cleared with 50% slurry of Protein A-Sepharose (Cytiva 17-0780-01; Merck) with the lysis buffer, then incubated with primary antibodies or IgG (sc-2025; Santa Cruz Biotechnology) for overnight at 4 °C with rotation, followed by incubation for 2 h with 50% slurry of Protein A-Sepharose presaturated with the lysis buffer. After washing three times with the lysis buffer, the immunoprecipitated proteins were recovered from the beads by boiling in the sample buffer with SDS and subject to Western blotting analysis.

Antibodies against RNF20 (ab32629; Abcam), UCP1 (ab10983; Abcam) TUBULIN (T6199; Sigma-Aldrich), TOMM20 (612272; BD Biosciences), SDHA (CSB-PA01985A0Rb; Cusabio), PPARγ (sc-7196; Santa Cruz Biotechnology), PLIN1 (20R-PP004; Fitzgerald), MYC-tag (05-724, Millipore), FLAG-tag (F1804; Sigma-Aldrich), HA-tag (3724; Cell Signalling Technology), NCoR1 (ab3482; Abcam), adiponectin (2789; Cell Signalling Technology), β-ACTIN (A5316; Sigma-Aldrich), GABPA (MA5-15419; Invitrogen), Mono- and polyubiquitinylated con-jugates (ENZ-ABS840; Enzo Life Sciences), GSK3β (610201; BD bioscience), phospho-GSK3β (9336; Cell Signalling Technology), VIN-CULIN (4650; Cell Signalling Technology), IgG (sc-2025; Santa Cruz Biotechnology) and OXPHOS (ab110413; Abcam) were used. The bands were visualised using horseradish peroxidase-conjugated secondary anti-rabbit or anti-mouse IgG antibodies (A0545 and A9044, respec-tively; Sigma-Aldrich). Uncropped images are provided in Source Data File.

For Western blotting quantification, we used ImageJ program to subtract the background from each blot (Process and Subtract Back-ground tools with a rolling ball radius of 300 pixels)[61]. We defined a rectangle that encompassed 30% of the width of each band, main-taining consistent size of these rectangles within the same blot, and then, the intensity of the rectangle was measured using ImageJ (Ana-lyse and Measure tools).

### Cell and whole-mount tissue imaging
Cells and whole or minced BAT and iWAT were washed with phosphate-buffered saline (PBS) and stained with Hoechst33342, BODIPY, MitoTracker™ Green, MitoTracker™ Deep-Red, or Lipid-TOX™ Deep-Red for 5 min. The samples were observed using a CQ1 confocal microscope (Yokogawa) and a coherent anti-Stokes Raman scattering microscope (CARS) (TCS SP8 CARS microscope, Leica Microsystems).

### Proteomic analysis
Proteins were extracted from BAT of WT and *Rnf20*+/− mice using RIPA buffer. Protein extracts were acetone-precipitated overnight at −20 °C, following which a tandem mass tag (TMT) was performed (AB Sciex, Framingham, MA, USA). Raw spectra were acquired using Orbitrap Fusion™ Lumos™ (Thermo Fisher) and EASY-nLC™ 1200 (Thermo Fisher). Raw MS spectra were processed using MaxQuant software (v1.5.8.3) at default settings with the UniProt *Mus musculus* database (16,987 reviewed protein sequences). Output files generated by Max-Quant were subjected to Perseus software (Proteome Software Inc.) to quantify TMT peptide and protein identification. Proteomics data was provided in Supplementary Data 1.

### Gene regulatory network analysis
The gene regulatory network was generated based on the transcription factor (TF) binding information from the ENCODE TF ChIP-seq (2015) and ChEA (2016) datasets retrieved from the Enrichr gene-set library[37]. Subnetworks of up- and down-regulated differentially expressed genes (DEGs) were induced from the gene regulatory network, including only DEGs and TFs that bind to the DEGs. DEGs with a fold change >1.5 and a *p*-value cutoff <0.1 in the MaxQuant software analysis were selected. Additional edges of the subnetworks were added to visualise interac-tions between genes, integrating the protein-protein interaction net-work from the STRING database v11.5[38]. TFs in the subnetworks were ranked by the *p*-value of the one-sided Fisher exact test with SciPy library v1.4.1 (scipy.stats.fisher_exact) using a contingency table of two variables: TF target genes and DEGs. Network plots were generated using Cytoscape v3.8.2 software[62].

### Luciferase assay
The luciferase assay was performed with a modified pGL4.24[luc2P/minP] enhancer luciferase vector system, in which the enhancer activity can be measured in the presence of minimal promoter sequences. DNA fragment of the mouse *Ucp1* (−2.5 kb upstream) enhancer (301 bp length) containing GABPα binding sites was PCR-amplified from C57BL/6 genomic DNA and cloned into the modified pGL3 basic luciferase vector. Also, the minimal promoter region of the pGL4.24 luciferase vector sequence was cloned and ligated between the mouse *Ucp1* enhancer region and β-galactosidase (primer sequences in Supplementary Data 1). The mouse *Pgc1α* (−38 kb upstream of the transcription start site) enhancer (339 bp length) containing GABPα binding sites was PCR-amplified from C57BL/6 genomic DNA using the primers listed in Supplementary Data 1 and

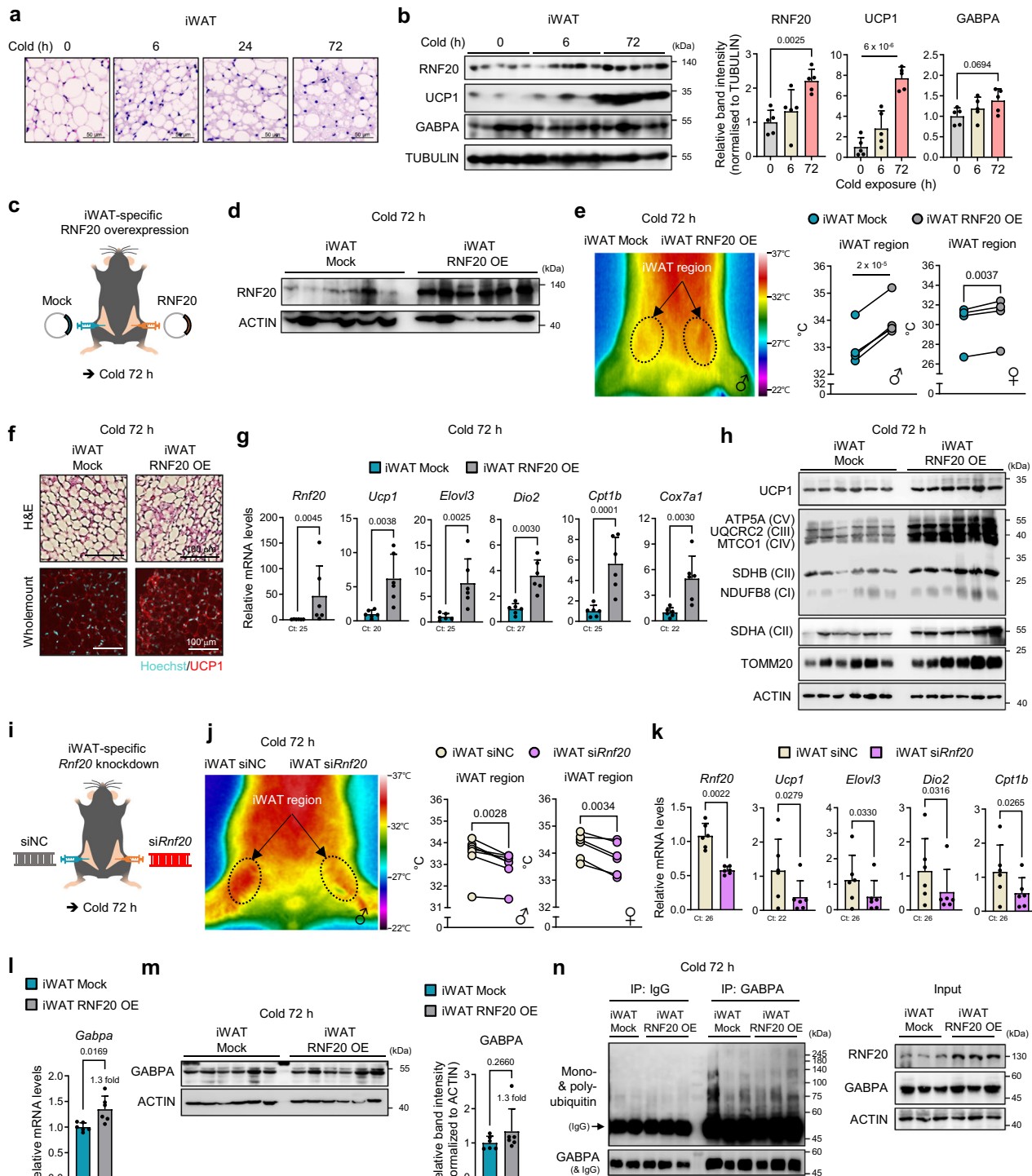

**Fig. 5 | In iWAT, RNF20 Potentiates Beige Fat Thermogenesis upon Prolonged Cold Stimuli, Independent of GABPα. a** Representative H&E staining of inguinal white adipose tissue (iWAT) from mice exposed to a cold environment (6 °C) for 6, 24, and 72 h. Scale bar: 50 μm. Representative results from two independent experiments. **b** Protein levels of RNF20 and UCP1 in iWAT during cold exposure $n = 5$ (RT), $n = 5$ (6 h), $n = 5$ (72 h) mice. Representative results from two independent experiments. **c** Experimental scheme for iWAT-specific RNF20 overexpression (5 μg). **d**–**h** RNF20 protein level, representative infrared image, iWAT region temperatures ($n = 5$ for male and $n = 4$ for female mice), representative H&E staining, qRT-PCR ($n = 6$ mice), and western blotting analysis of iWAT from mock and RNF20 OE mice exposed to cold (6 °C) for 72 h. Representative results from two

independent experiments. **i** Experimental scheme for iWAT-specific siRNA treatment (5 μg). **j, k** iWAT region temperatures ($n = 7$ for male and $n = 6$ for female mice) and qRT-PCR analysis ($n = 6$ mice) of iWAT from siNC and si*Rnf20* mice exposed to cold (6 °C) for 72 h. **l, m** GABPα mRNA and protein levels in iWAT from iWAT RNF20 OE mice exposed to cold (6 °C) for 72 h. **n** Ubiquitin levels of GABPα in iWAT from mock and RNF20 OE mice. Representative results from two independent experiments. Source data are provided as a Source Data file. $n$ indicates biological replicates. Data are represented as mean ± S.D. Significance was determined using one-way ANOVA with Dunnett's multiple comparison (**b**) and paired two-sided Student's $t$-test (**e, g, j, k, l, m**).

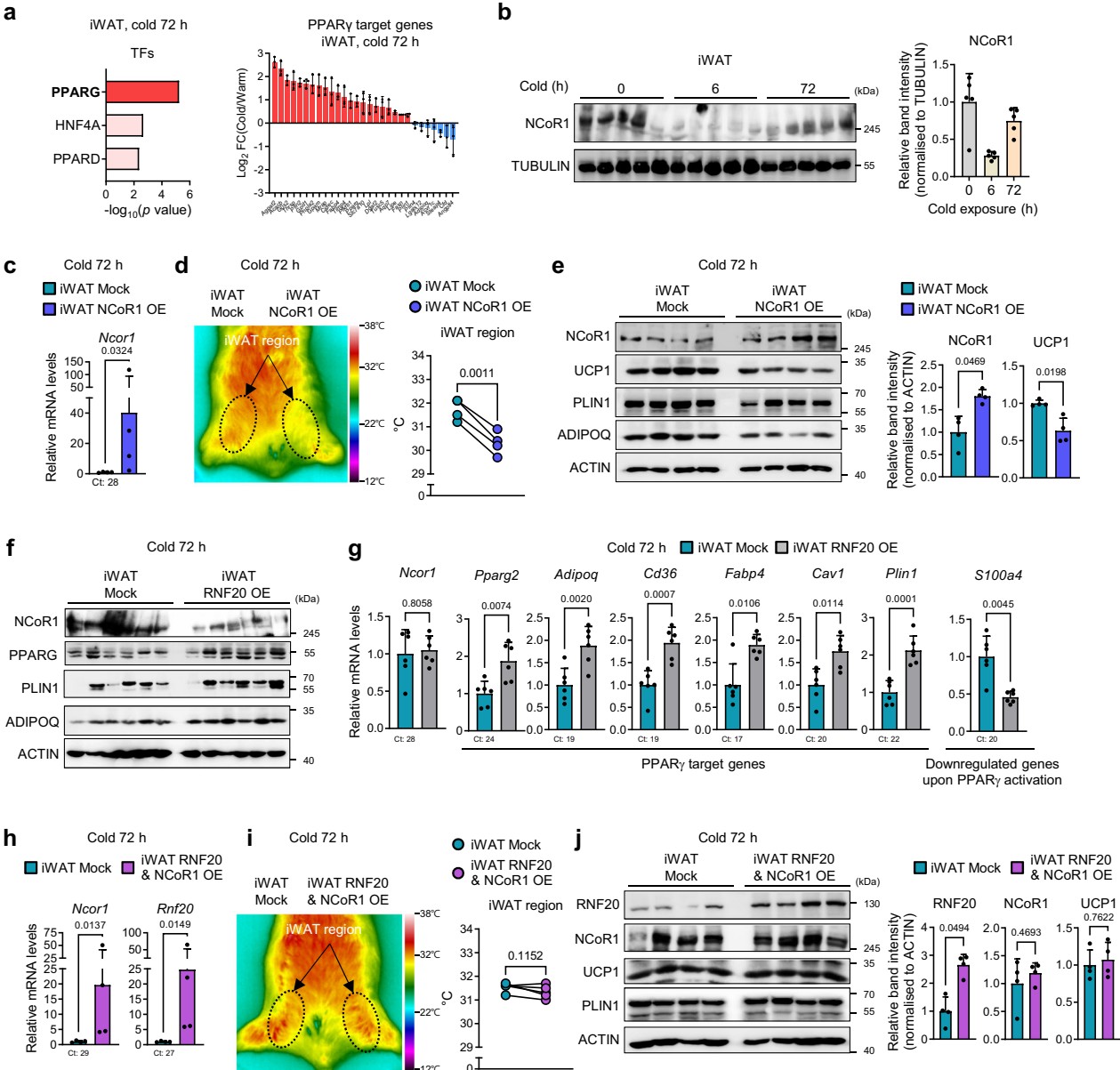

**Fig. 6 | iWAT RNF20 Potentiates Beige Fat Thermogenesis by Promoting NCoR1 Degradation to Stimulate PPARγ. a** Gene ontology and transcription factor analysis of upregulated genes in iWAT transcriptome and expression levels of PPARγ target genes in iWAT upon 3-day cold stimuli (6 °C, GSE179385). **b** Protein levels of NCoR1 in iWAT during cold exposure (6 °C). $n = 5$ (RT), $n = 5$ (6 h), $n = 5$ (72 h) mice. Representative results from two independent experiments. **c**–**e** *Ncor1* mRNA expression level, representative infrared image, iWAT region temperatures ($n = 4$ mice), and western blot analyses of iWAT from mock and NCoR1 OE mice (5 μg) exposed to cold (6 °C) for 3 d. **f, g** Western blot analysis and qRT-PCR ($n = 6$ mice) of iWAT from mock and RNF20 OE mice exposed to cold (6 °C) for 3 days. **h**–**j** *Ncor1* and *Rnf20* mRNA expression levels, representative infrared image, iWAT region temperatures ($n = 4$ mice), and western blotting analyses of iWAT from mock and RNF20&NCoR1 OE mice (5 μg for each gene) exposed to cold (6 °C) for 3 d. Source data are provided as a Source Data file. $n$ indicates biological replicates. Data are represented as the mean ± S.D. Significance was determined using Fisher's exact test (**a**), unpaired two-sided Student's *t*-test (**a**) and paired two-sided Student's *t*-test (**c**–**e**, **g**–**j**).

cloned between the KpnI and XhoI sites of the modified pGL3 basic luciferase vector.

For the mouse *Mtif2* promoter luciferase assay, the mouse *Mtif2* promoter (−86 bp upstream of the transcription start site) sequence (105 bp length) was amplified with XhoI and HindIII and cloned into the pGL3 basic luciferase system vector (cloning primer sequences in Supplementary Data 1). Luciferase assays were performed using transient transfection of HEK293T cells using the calcium phosphate transfection method[63]. 24 h after transfection, the cell lysates were harvested and extracted using lysis buffer [25 mM Tris-phosphate (pH 7.8), 10% glycerol, 2 mM EDTA, 2 mM dithiothreitol, and 1% Triton™ X-

100], and luciferase and β-galactosidase activities were measured according to the manufacturer's protocol (E1500, Promega). The relative luminescence units were normalised to β-galactosidase activity.

**Transient transfection**

siRNAs for mouse *Rnf20, Gabpa*, human *RNF20*, and the negative control were produced by Bioneer, Inc. (Daejeon, South Korea). Pre-adipocytes or differentiated adipocytes were mixed with siRNA or vectors using the Neon™ Transfection Kit (MPK1096 and MPK10096, Thermo Fisher) and transfected with a single pulse at 1100 V for 30 ms

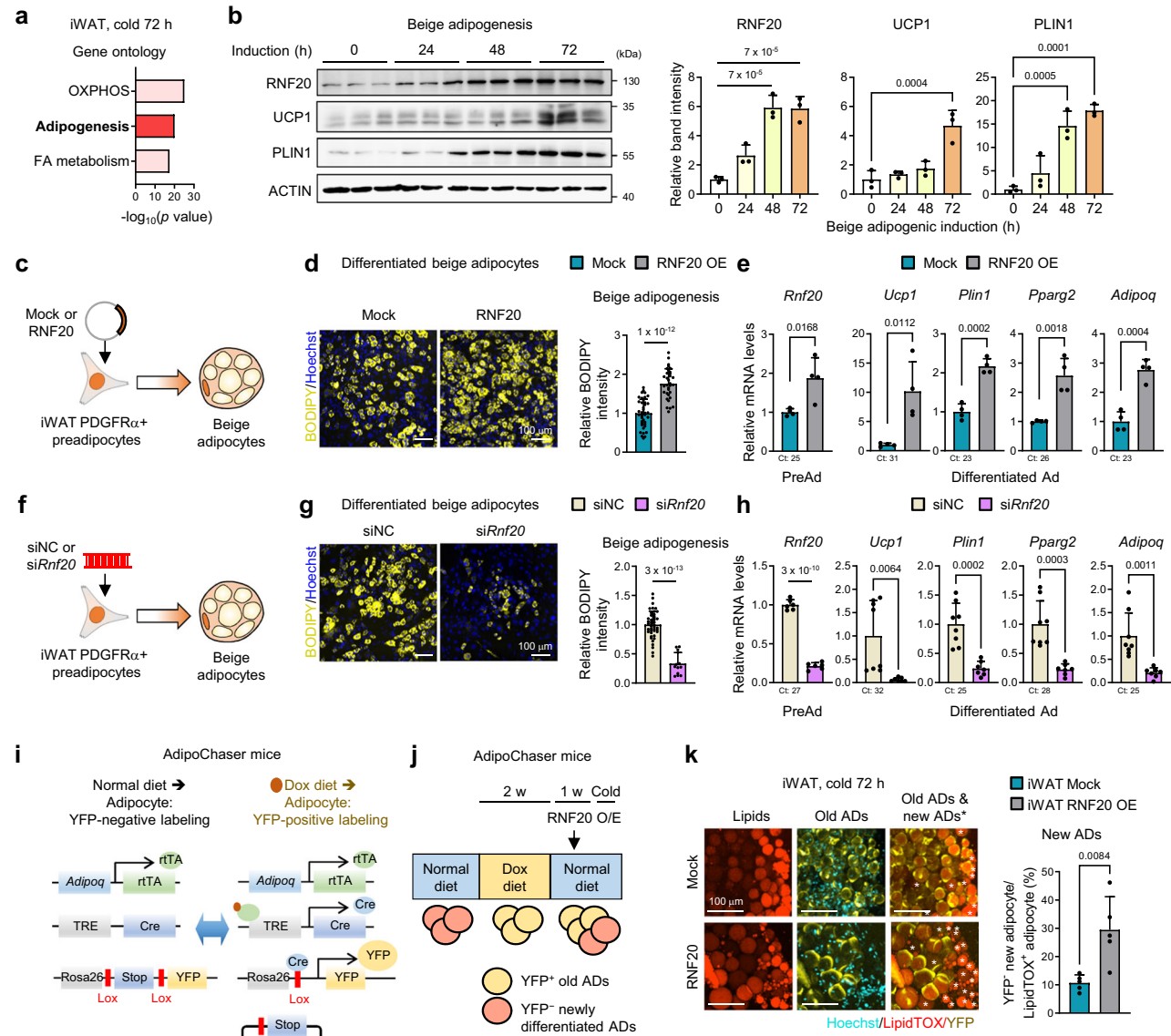

**Fig. 7 | iWAT RNF20 potentiates de novo beige adipogenesis upon prolonged cold stimuli. a** Gene ontology of upregulated genes in iWAT transcriptome upon 3-day cold stimuli (6 °C, GSE179385). **b** Western blot analysis of differentiated beige adipocytes from iWAT PDGFRα-expressing preadipocytes n = 3 (0 h), n = 3 (24 h), n = 3 (48 h), n = 3 (72 h) independent replicates. Representative results from two independent experiments. **c–h** Experimental scheme, a representative image of BODIPY lipid staining and qRT-PCR analysis of differentiated beige adipocytes from iWAT PDGFRα-expressing preadipocytes transduced with mock or RNF20-expressing plasmids (**c–e**) and siNC or si*Rnf20* (**f–h**) before adipogenic induction. **i, j** Experimental scheme using AdipoChaser mice. After 2 weeks of a diet containing

doxycycline for YFP labelling, a normal diet was fed. After injection of mock or RNF20 into the left or right iWAT flank (5 μg), respectively, AdipoChaser mice were exposed to cold (6 °C). **k** Representative whole-mount images of iWAT from AdipoChaser mice upon cold exposure (6 °C) for 3 d. The asterisk (*) refers to newly differentiated YFP-negative adipocytes. AD adipocytes. Source data are provided as a Source Data file. *n* indicates biological replicates. Data are represented as mean ± S.D. Significance was determined using Fisher's exact test (**a**), one-way ANOVA with Dunnett's multiple comparison (**b**) and unpaired two-sided Student's *t*-test (**d, e, g, h, k**).

---

using a Microporator MP-100 (Digital Bio, Seoul, Korea). HEK293T cells were transfected using Lipofectamine™ RNAiMAX (13778075, Thermo Fisher) according to the manufacturer's protocol or the calcium phosphate method. The sequence information of the siRNAs is described in Supplementary Data 1.

**Chromatin immunoprecipitation assay (ChIP)**
BAT was minced directly into a cross-linking solution (10 ml, 1% formaldehyde diluted in PBS) for 20 min at RT[64]. Cross-linking was quenched by adding 0.5 ml of 2.5 M glycine for 5 min, followed by three washes with ice-cold PBS. SDS lysis buffer [50 mM Tris-HCl (pH 8.1), 10 mM EDTA, 1% SDS, and protease inhibitor (PI)] was then added

to the samples. Chromatin fragmentation was performed using probe sonication at 4 °C for six cycles of 10 s at middle amplitude, with a 30 s pause on ice between cycles. Chromatin lysates were diluted with nine volumes of ChIP dilution buffer (16.7 mM Tris-HCl pH 8.1, 1.2 mM EDTA, 0.01% SDS, 167 mM NaCl, 1.1% Triton™ X-100, and PI). After centrifugation, the input was saved, and lysates were incubated with antibodies overnight at 4 °C and then immunoprecipitated with Protein A Sepharose CL-4B beads with rotation for 2 h at 4 °C. Immunoprecipitation was performed as follows: one wash with 1 ml of low-salt wash buffer (20 mM Tris-HCl pH 8.1, 2 mM EDTA, 0.1% SDS, 150 mM NaCl, 1% Triton™ X-100, and PI), one wash with 1 ml high salt buffer (20 mM Tris-HCl pH 8.1, 2 mM EDTA, 0.1% SDS, 500 mM NaCl, 1%

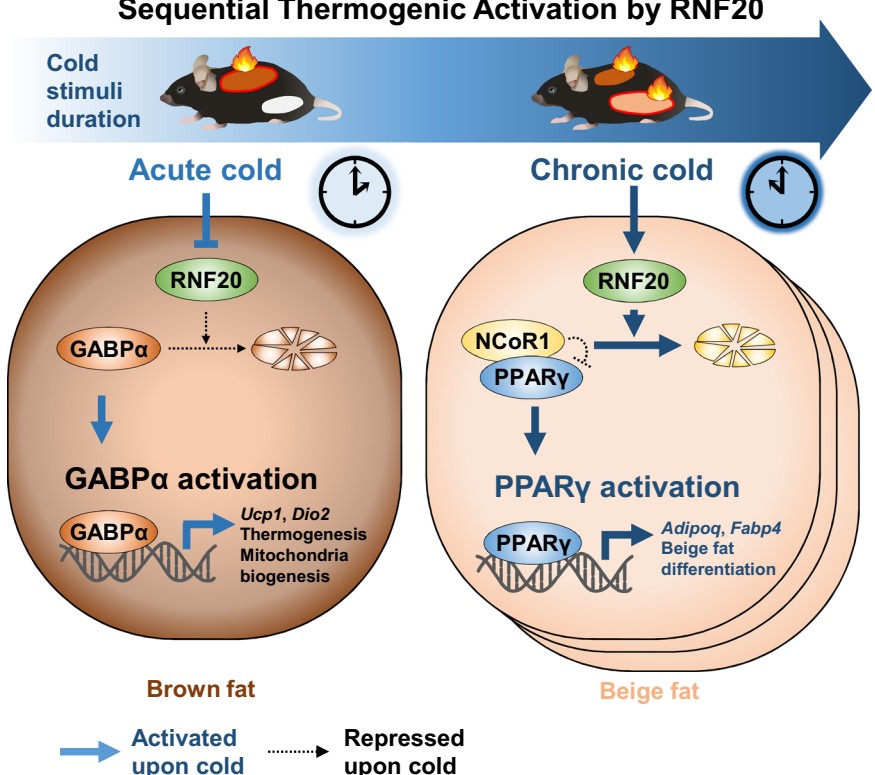

**Fig. 8 | Proposed model.** Upon cold stimuli, mammals maintain their body temperature by potentiating adipose thermogenesis in a fat-depot-specific manner. In BAT, acute cold stimuli rapidly downregulate RNF20, which induces GABPα accumulation and activation. In turn, GABPα enhances thermogenesis by stimulating the expression of thermogenic and mitochondrial genes. Given that RNF20-NCoR1 axis in BAT would primarily regulate lipid metabolism, NCoR1 in BAT is not marked. In iWAT, prolonged cold stimuli gradually upregulate RNF20 to stimulate PPARγ by degrading NCoR1, which promotes de novo beige adipogenesis.

Triton™ X-100, and PI), three washes with 1 ml LiCl buffer (10 mM Tris-HCl pH 8.1, 1 mM EDTA, 250 mM LiCl, 1% deoxycholate, and PI), and two washes with 1 ml TE buffer (10 mM Tris-HCl pH 8.1, 1 mM EDTA, and PI). Lysates were eluted twice with 250 µl elution buffer (100 mM NaHCO₃ and 1% SDS), at 65 °C for 15 min, with rotation, following which the elutes were combined (total 500 µl). Cross-linking was reversed overnight at 65 °C with 20 µl of 5 M NaCl (final concentration: 200 mM NaCl), then incubated with proteinase K, and DNA was isolated using phenol/chloroform extraction, followed by ethanol DNA precipitation. Antibodies against GABPα (MA5-15419; Invitrogen) and mouse IgG (A9044; Sigma-Aldrich) were used. The sequence information for ChIP is presented in Supplementary Data 1.

### Estimation of mitochondrial DNA (mtDNA) copy number
Approximately 10–20 mg of BAT was homogenised in 500 µl of lysis buffer (50 mM Tris pH 8.0, 10 mM EDTA pH 8.0, 100 mM NaCl, and 2% SDS). The samples were then incubated overnight at 55 °C with proteinase K (50 µg/sample). After incubation with RNase A (100 µg/sample) for 3 h at 37 °C, the DNA was precipitated using ethanol. The DNA samples were diluted to a concentration of 10 ng/µl and subjected to qRT-PCR analysis using mt-ND4L and *Hk2* primers to amplify mtDNA and nuclear DNA (nDNA), respectively. Sequence information for qRT-PCR is presented in Supplementary Data 1.

### Cellular oxygen consumption assay
Cellular OCR was analysed using a Seahorse XFe96 extracellular flux analyser (Agilent) according to the manufacturer's instructions. Differentiated brown adipocytes were incubated in an assay medium (25 mM glucose, 1 mM sodium pyruvate, 2 mM L-glutamine, and 1% fatty acid-free bovine saline albumin in Seahorse XF base medium at

pH 7.4). Cell numbers were determined by Hoechst staining and used to normalise the OCRs.

### Indirect calorimetry
Indirect calorimetry was performed using PhenoMaster (TSE Systems) according to the manufacturer's protocol. Male mice, aged 3–4 months, were placed in a calorimetric chamber. To activate β3-adrenergic signalling, mice were intraperitoneally injected with CL-316,243 (0.5 mg/kg).

### Public single-cell RNA-seq data analysis
We analysed gene-by-cell count matrix in the public dataset GSE125269 (single-cell RNA-seq of brown adipocytes). In the analysis of dataset GSE125269, cells with >75% unique molecular identifiers (UMIs) assigned to mitochondrial genes, <5000 total UMI counts, or <500 detected genes were excluded. Clustering was performed using the FindClusters function of the Seurat (v3.0.2) R package[65] and visualised by a t-SNE plot using the RunTSNE function of the same package. Adipocyte clusters were identified by *Adipoq* expression and annotated to high thermogenic adipocytes, low thermogenic adipocytes, and white adipocytes based on the expression levels of *Ucp1* and thermogenic genes. Gene expression of adipocyte clusters was visualised using the VlnPlot function, and differentially expressed genes between low and high thermogenic adipocytes were identified using the FindMarkers function of the Seurat package.

### qRT-PCR
Total RNA was isolated from tissues or cells using TRIzol Reagent (RiboEx, GeneAll) or Direct-zol™ RNA MiniPrep (Zymo Research) for small amounts of RNA extraction), and subjected to cDNA synthesis

using the ReverTra Ace qPCR RT Kit (Toyobo). Relative mRNA levels were detected using the CFX96TM Real-Time System (Bio-Rad Laboratories). qRT-PCR was performed using SYBR Green Master Mix (DQ384-40h, Biofact). Target gene expression levels were normalised to cyclophilin gene expression levels. The sequence information for qRT-PCR is described in Supplementary Data 1.

## Plasmid information

Myc-tagged RNF20 was constructed using the pcDNA3.1(+) vector[25]. Flag-tagged NCoR1 was constructed using the pCMX vector, which was kindly provided by Sung Hee Baek from Seoul National University. For Flag-tagged GABPα and GABPβ1, the p3xFLAG-CMV-10 vector was used (cloning primer sequences Supplementary Data 1).

## Insulin and glucose tolerance test

For the insulin tolerance test (ITT), mice were fasted for 2 h and insulin was administered (0.75 units/kg body weight, 91077 C; Sigma-Aldrich, St. Louis, MO). In the Extended Fig. 7n experiment, mice were fed *ad libitum* and insulin was administered (0.5 units/kg body weight) to avoid hypoglycemic shock. For the intraperitoneal glucose tolerance test (GTT), mice were fasted for 6 h and glucose was administered (2 g/kg body weight). For the insulin signalling examination, mice were fasted for 2 h and insulin was administered.

## Statistical analysis

Data are presented as the mean ± standard deviation. In each figure, the number of biological replicates for each experiment (*n*) was dictated. The number of independent experiments and relevant statistical methods for each panel were described in the figure legends. All data were tested for normal distribution using the D'Agostino-Pearson omnibus normality test. If the data were not normally distributed, the Mann–Whitney *U* test was performed. The means of the two groups were compared using two-tailed Student's *t*-test. The means of multiple groups were compared using one-way analysis of variance (ANOVA), followed by Dunnett's or Tukey's multiple comparisons test. Two independent variables were compared using two-way ANOVA, followed by Sidak's multiple comparisons test. Statistical analyses were performed using Prism v. 10.0.2 (GraphPad Software).

## Reporting summary

Further information on research design is available in the Nature Portfolio Reporting Summary linked to this article.

# Data availability

Proteomics data are provided in Supplementary Data 1. The used public data were GSE119452[66] for BAT RNA-seq, GSE125269[34] for BAT scRNA-seq, GSE63964[67] for BAT ChIP-seq, GSE68544[68] for brown adipocyte microarray, GSE179385[21] for iWAT RNA-seq, GSE98132[69] for WAT and BAT RNA-seq, and human adipose tissue proteome[70] [https://static-content.springer.com/esm/art%3A10.1038%2Fsrep30030/MediaObjects/41598_2016_BFsrep30030_MOESM2_ESM.xls]. The mass spectrometry proteomics data have been deposited to the ProteomeXchange Consortium via the PRIDE[71] partner repository with the dataset identifier PXD048556. Source data are provided with this paper.

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

## Acknowledgements

We thank Dr. Kai Ge at the National Institutes of Health for kindly pro-viding the immortalised brown preadipocyte cell line. We also appreciate the Korea Mouse Phenotyping Centre for analysing the

metabolic phenotypes of the mice. This study was supported by the National Research Foundation, funded by the Korean government (Ministry of Science and ICT; NRF-2020R1A3B2078617, NRF-2018R1A5A1024340, and RS-2023-00218616 to J.B.K., and NRF-2022R1C1C2003113 to Y.G.J.). N.H., Y.J., J.H.S., J.S.H., S.M.K., W.T.L., and Jeu. P. were supported by the BK21 Plus programme.

## Author contributions

Y.G.J. conceptualised the study, performed the experiments, analysed the data, and wrote the manuscript. H.N. performed the mouse experiments and analysed the scRNA-seq data. J.O. and Jiyoung. P. performed AdipoChaser mouse experiments. D.L., D.W.K., and J.-Y.C. performed and analysed proteomics experiments. J.E.K., Y.Y.K., Y.J., J.S.H., J.H.S., S.M.K., W.T.L., S.W.K., Jeu P., and J.Y.H. performed experiments. K.J. analysed proteomics data. J.B.K. supervised the study and wrote the manuscript.

## Competing interests

The authors declare no competing interests.
