## [Peer Review File · Nature Communications]

Ubiquitin Ligase RNF20 Coordinates Sequential Adipose Thermogenesis with Brown and Beige Fat-Specific SubstratesEditorial Note: Parts of this Peer Review File have been redacted as indicated to remove third-party material where no permission to publish could be obtained.

REVIEWER COMMENTS

Reviewer #1 (Remarks to the Author):

This manuscript identified RNF20, an E3 ubiquitin ligase, as an important regulator of thermogenesis, which plays distinct roles in brown and beige fat cells. In BAT, the authors found that the RNF20 protein level is negatively correlated with UCP1 protein level.

Manipulating RNF20 expression in vivo significantly alters the thermogenesis capacity of BAT. Proteomics study identified the transcription factor GABPalpha as a target of RNF20. GABPalpha knockdown reduced UCP1 expression and diminished the effect of RNF20 reduction induced elevation in UCP1 expression. Biochemistry experiments validated that RNF20 targets GABPalpha for degradation. Interestingly, unlike in BAT, the RNF20 protein level in iWAT is positively correlated with UCP1 protein level. In iWAT, RNF20 targets NCoR1 for degradation, which stimulates PPARgamma expression to promote de novo beige adipogenesis after prolonged exposure.

Brown and beige adipocytes behave differently in response to cold stimulation to activate thermogenesis. Brown adipocytes in BAT are considered to be always there, and ready to be activated, and beige adipocytes in iWAT require continued cold stimulation to activate de novo beige adipogenesis. This is a timely study that identifies a novel molecular mechanism that is oppositely required for BAT and iWAT to activate thermogenesis. The authors conducted the study with a highly organized, logical, and sophisticated experimental design, providing ample data to substantiate their major conclusions. I only have a few suggestions that could potentially improve the manuscript.

1. It would be nice to separate the role of RNF20 in adipose progenitor cells and mature beige adipocytes. For example, what cell type is NCoR1 enriched in iWAT? Is NCoR1 highly enriched in the progenitor cells to drive beige adipogenesis? What is the role of RNF20 in mature beige adipocytes? Does RNF20 play a similar role regarding thermogenesis in mature beige and brown adipocytes?
2. How about NCoR1 protein expression in BAT after short-term and prolonged cold exposure when there is potentially brown adipogenesis?

Reviewer #2 (Remarks to the Author):

NCOMMS-23-09785-T

Ubiquitin Ligase RNF20 Coordinates Sequential Adipose Thermogenesis with Brown and Beige Fat-Specific Substrates

Thermogenic activation triggers an intricate molecular process in the adipose tissue by activating and inhibiting the right set of genes and proteins. Ubiquitination facilitates rapid degradation of the proteins that are not required or otherwise inhibit thermogenesis. Jeon et al's manuscript provides detailed analyses of how the E3 ubiquitin ligase, RNF20, could potentiate thermogenesis in the brown and white adipose tissue depots in mice. While the brown adipose data clearly demonstrates that the loss of Rnf20 promotes thermogenesis by preventing GABPa degradation, the white adipose tissue data seems both interesting and contradictory to the group's previous publication (PMID: 31604693).

Major comments

1. Given that the Rnf20 KO mice are insulin sensitive and have enhanced BAT activity during cold exposure, adipose-specific Rnf20 KO or OE mice could be better experimental models to understand the role of Rnf20 in thermogenesis better.
2. Since Rnf20 promotes ubiquitination in a cell type-specific manner (Line 20) and adipocyte progenitors are heterogenic in nature, in vitro experiments should have been performed in the isolated adipocyte progenitor cells (E.g., Fig 7c-h).
3. Fig 6 only shows that Rnf20 OE upregulates WAT markers (6f-g). Could the authors also show the gene/protein expression of thermogenic markers/fatty acid oxidation in the whole iWAT (a couple of transcription factors and downstream targets)?
4. Likewise, figs 7i-k could also benefit from the inclusion of thermogenic markers.
5. Did the authors measure the impact of iWAT Rnf20 OE on whole-body insulin sensitivity or glucose tolerance?
6. Are the plasmids specifically designed to target adipocyte progenitors or mature

adipocytes at the injection site? Plasmid information is unavailable in the methods.

Minor comments

1. Some samples have been labelled as “ct-27, ct29, ct 33, etc.” in some figures, which is confusing.
2. It is hard to understand the blot data (e.g., 2e, 5h, 5i, 1l, 6e, 6e, 7h). The authors could have used a higher % gel to separate the smaller kDa proteins.
3. Symbols used in the extended data need to be fixed.
4. Line 198, Mtif2 assay, is not written in detail in the methods/figure legend/results sections.
5. How 72 h cold exposure time point was adopted?

Reviewer #3 (Remarks to the Author):

The authors showed that RNF20 decreases during acute cold and enhances thermogenic activity in BAT, while RNF20 increases during chronic cold in iWAT. Furthermore, the authors showed that the regulation of thermogenesis in BAT is mediated by the accumulation of GABP α .

The RNF20-GABP α axis in BAT thermogenesis is a very important finding. However, the molecular mechanisms mediated by GABP α and NCoR1 and the function of RNF20 in iWAT are not sufficiently substantiated, as most of them are based on overexpression systems. Numerous additional key direct data are needed to support the main claims of the paper.

Major concern:

1. Adipocyte-specific KO mice should be used for analysis. Most of the data is due to OE, especially in experiments using iWAT, and at least KO and siRNA should be used in the analysis.

Fig.5: Please verify whether thermogenic activity is enhanced during Chronic cold in iWAT.

Fig.6: How about in RNF20KO/KD and NCoR1KO/KD conditions?

Fig.7i-k: How about in RNF20KO/KD conditions?

2. Fig.1a, 1b, 4n, 5b, 6b, 7b Extended Data Fig.2a: The analysis should be performed with

N \geq 3.

3. Fig.2g, 2i, 2o, 2q, 3o: The BAT Mock and BAT siNC data should be equivalent if they are measured under the same conditions. If they are analyzed under different conditions, they should be clearly indicated, and if not, they should be reanalyzed under more rigorous conditions or with a greater number of analyses.

4. It is not certain whether GABP α undergoes ubiquitin-dependent degradation in BAT. The band of ubiquitination in Fig. 4k does not appear to be the typical smear/ladder-like band. It is not possible to evaluate this ubiquitination assay because the experimental conditions are not described, but when detecting by ubiquitin antibody, proteins that co-immunoprecipitate must be eliminated by denaturing conditions or other means to be evaluated correctly. That is, it may be overestimated due to contamination of ubiquitination of the binding proteins. One method is to perform a pull-down with TUBE (Tandem ubiquitin binding entity) and evaluate it by substrate (i.e. GABP α) antibody. When performing under RNF20OE conditions, please add RNF20 deltaRING mutant, and also perform under KO/KD conditions.

5. Regarding BAT-specific decrease of GABP α , authors points to differences in expression levels as one possibility. Does overexpression of GABP α in iWAT under chronic cold conditions lead to GABP α degradation?

6. Further investigation is needed to determine whether GABP α undergoes degradation in iWAT or not. There is a possibility that synthesis and degradation are balanced. First, for degradation, please perform the same experiment as Fig. 4n under normal and cold conditions with RNF20 \pm BAT and iWAT. Please measure Gabp α mRNA levels in iWAT cells under normal, Chronic cold, and beige adipogenesis conditions with respect to synthesis.

7. Ubiquitin-dependent degradation of NCoR1 has been partially implicated in HEK293T and 3T3-L1 cells in a previous report by the authors (ref. 25). In line with the author's claim that RNF20 ubiquitin-dependent degradation of GABP α only in specific cell populations, the ubiquitin-dependent degradation of NCoR1 must be proven in each cell type. Since the

authors claim that NCoR1 is a substrate of RNF20 in iWAT, please prove ubiquitination-dependent degradation, at least in this iWAT. In addition, how do NCoR1 protein and mRNA levels change in iWAT RNF20KO/KD under normal, chronic cold, and beige adipogenesis conditions?

Line324-326: I may have missed it, but I could not identify any data where NCoR1 undergoes ubiquitination-dependent degradation in BAT, at least in refs. 25 and 26. Is there any other evidence? How do NCoR1 protein and mRNA levels change under normal and acute cold conditions in BAT? And does it vary by RNF20KO/KD?

8. Fig.1k, 1l: Ucp1 mRNA is increased about 1.3-fold by BAT RNF20 +/-, but UCP1 protein appears to be increased more than 2-fold.

Fig.2d, 2e: Ucp1 mRNA is reduced about half by BAT RNF20 OE, but UCP1 protein is almost lost. Ucp1 seems to be under control in the process of translation and beyond, how would the author explain it?

Minor comments:

Fig.2: Authors should perform qPCR of Ucp1 and Pgc1a under siRNF20 or RNF20+/- conditions.

Fig.3p: Authors should perform qPCR of Ucp1 and Gabpa under siRNF20 or RNF20+/- conditions.

Fig.5b, 6b: Authors should perform qPCR of RNF20, Ucp1, Gabpa, and NCoR1 under iWAT/chronic cold condition.

Fig.7b: Authors should perform qPCR of RNF20, Ucp1, Gabpa, and PLIN1 under iWAT/beige adipogenesis condition.

Fig.4j, 4k: Please add IB: GABPA

Fig.7h: Please add IB: RNF20

Reviewers' Comments:

Reviewer #1 (Remarks to the Author):

This manuscript identified RNF20, an E3 ubiquitin ligase, as an important regulator of thermogenesis, which plays distinct roles in brown and beige fat cells. In BAT, the authors found that the RNF20 protein level is negatively correlated with UCP1 protein level. Manipulating RNF20 expression in vivo significantly alters the thermogenesis capacity of BAT. Proteomics study identified the transcription factor GABPalpha as a target of RNF20. GABPalpha knockdown reduced UCP1 expression and diminished the effect of RNF20 reduction induced elevation in UCP1 expression. Biochemistry experiments validated that RNF20 targets GABPalpha for degradation. Interestingly, unlike in BAT, the RNF20 protein level in iWAT is positively correlated with UCP1 protein level. In iWAT, RNF20 targets NCoR1 for degradation, which stimulates PPARgamma expression to promote de novo beige adipogenesis after prolonged exposure.

Brown and beige adipocytes behave differently in response to cold stimulation to activate thermogenesis. Brown adipocytes in BAT are considered to be always there, and ready to be activated, and beige adipocytes in iWAT require continued cold stimulation to activate de novo beige adipogenesis. This is a timely study that identifies a novel molecular mechanism that is oppositely required for BAT and iWAT to activate thermogenesis. The authors conducted the study with a highly organized, logical, and sophisticated experimental design, providing ample data to substantiate their major conclusions. I only have a few suggestions that could potentially improve the manuscript.

1. It would be nice to separate the role of RNF20 in adipose progenitor cells and mature beige adipocytes. For example, what cell type is NCoR1 enriched in iWAT? Is NCoR1 highly enriched in the progenitor cells to drive beige adipogenesis? What is the role of RNF20 in mature beige adipocytes? Does RNF20 play a similar role regarding thermogenesis in mature beige and brown adipocytes?

Thanks for these intriguing comments. To investigate which cell types might abundantly express *Ncor1* in WAT, we examined recently published single-nucleus RNA-sequencing of WAT^{1,2}. Our analyses revealed that *Ncor1* is expressed across all cell types including preadipocytes (adipose stem and progenitor cells, ASPC and fibro-adipogenic progenitors, FAPs) and adipocytes in mouse and human WATs (**Reviewer's only Fig. 1a–c**), implying that NCoR1 appears not to be enriched certain cell types including adipose progenitor cells nor mature adipocytes. Further, when we examined mRNA expression levels of *Ncor1* in adipose stem cells³⁻⁵, *Ncor1* is expressed all adipose stem cell subpopulations including BST2^{high} beige precursors (I4 cluster in **Reviewer's only Fig. 1d**), *Icam1*+ adipocyte precursor cells (*Icam1*+ APCs in **Reviewer's only Fig. 1e**), and PPARG⁺ adipocyte precursor cells (APCs in **Reviewer's only Fig. 1f**). Thus, it is likely that the NCoR1 in WAT might primarily affect *de novo* adipogenesis rather than beiging process in iWAT.

Next, we investigated the role of RNF20 separately in preadipocytes and mature beige adipocytes. To address this, we modulated RNF20 expression in iWAT CD31⁻CD45⁻PDGFR α ⁺ preadipocytes⁶ and then differentiated them into beige adipocytes. Our findings revealed that RNF20 overexpression (OE) upregulated lipid accumulation and mRNA expression of adipogenic and thermogenic genes, whereas RNF20 knockdown (KD) downregulated these phenomena (**new Fig. 7c–h and Extended Data Fig. 10c–e**), indicating that RNF20 would contribute to potentiating beige adipogenesis in progenitor cells of iWAT.

Further, we examined the role of RNF20 in mature beige adipocytes. When we modulated RNF20 expression in mature beige adipocytes derived from iWAT CD31⁻CD45⁻PDGFR α ⁺ preadipocytes, RNF20 OE potentiated the expression levels of thermogenic genes and PPAR γ target genes whereas RNF20 KD repressed these (**new Extended Data Fig. 10f–h**), which is consistent with the effect of RNF20 on PPAR γ activation. We described these in the revised manuscript (pp. 12 and 13).

In this study, we propose that the role, regulation, and primary substrates of RNF20 appear to be different in brown adipocytes and beige precursor/adipocytes, which is crucial for coordinating sequential thermogenesis upon cold stimuli. In brown adipocytes, the primary substrate of RNF20 would be GABP α . Upon acute cold stimuli, RNF20 was rapidly decreased in brown adipocytes, which boosted thermogenesis by accumulating and activating GABP α . On the other hand, in beige precursors and beige adipocytes from iWAT, it seems that the primary substrate of RNF20 would be NCoR1. Upon prolonged cold stimuli, the protein level of RNF20 in iWAT was gradually increased, thereby enhancing *de novo* beige adipogenesis and thermogenic activity by promoting NCoR1 degradation and PPAR γ activation. Collectively, these data suggest that RNF20 would play an important role in the sequential activation of brown and beige fat thermogenesis with different substrate specificity during cold stimuli.

2. How about NCoR1 protein expression in BAT after short-term and prolonged cold exposure when there is potentially brown adipogenesis?

Per this comment, we examined NCoR1 protein levels in BAT after short-term (6 h) and prolonged cold exposure (72 h). As shown in **new Extended Data Fig. 2a**, the level of NCoR1 protein in BAT was upregulated upon acute (6 h) and prolonged (72 h) cold stimuli, which was inversely correlated with RNF20 protein level (**new Fig. 1b**). We described these in the revised manuscript (p. 5).

[EDITORIAL NOTE: Figure redacted]

Reviewer's only Fig. 1. mRNA expression levels of mouse and human *Ncor1* in single-nucleus and single-cell RNA sequencing data of white adipose tissue (Data from a, c: SCP1179; b: SCP1376, d: PRJNA708350, e: GSE128890, f: GSE144299).

Reviewer #2 (Remarks to the Author):

NCOMMS-23-09785-T

Ubiquitin Ligase RNF20 Coordinates Sequential Adipose Thermogenesis with Brown and Beige Fat-Specific Substrates

Thermogenic activation triggers an intricate molecular process in the adipose tissue by activating and inhibiting the right set of genes and proteins. Ubiquitination facilitates rapid degradation of the proteins that are not required or otherwise inhibit thermogenesis. Jeon et al's manuscript provides detailed analyses of how the E3 ubiquitin ligase, RNF20, could potentiate thermogenesis in the brown and white adipose tissue depots in mice. While the brown adipose data clearly demonstrates that the loss of Rnf20 promotes thermogenesis by preventing GABPa degradation, the white adipose tissue data seems both interesting and contradictory to the group's previous publication (PMID: 31604693).

Major comments

1. Given that the Rnf20 KO mice are insulin sensitive and have enhanced BAT activity during cold exposure, adipose-specific Rnf20 KO or OE mice could be better experimental models to understand the role of Rnf20 in thermogenesis better.

We fully recognize the significance of adipose-specific *Rnf20* KO or OE mouse model to enhance the value of our current findings. Despite our best efforts over several years, we have encountered challenges in generating adipocyte-specific or brown adipocyte-specific *Rnf20* deletion mouse models. As an alternative approach, we have modulated *Rnf20* expression specifically in BAT or iWAT through direct nucleotide injection and utilized cell line models. These approaches have been successful in investigating cell- or tissue-specific gene functions^{3,7-9}. Notwithstanding these approaches, we addressed the need for future studies generating and employing brown adipocyte-specific or beige precursor-specific *Rnf20* gain- and loss-of-function mouse models in the revised manuscript (p. 16).

2. Since Rnf20 promotes ubiquitination in a cell type-specific manner (Line 20) and adipocyte progenitors are heterogenic in nature, in vitro experiments should have been performed in the isolated adipocyte progenitor cells (e.g., Fig 7c-h).

Thanks for the suggestion. To address this, we performed *in vitro* cell culture experiments with CD31⁻CD45⁻PDGFR α -expressing preadipocytes, one of the cell types including beige precursors⁶, and differentiated them into mature beige adipocytes. As shown in new **Fig. 7b–h and Extended Data Fig. 10a–e**, our data suggested that RNF20 would promote beige adipogenesis of PDGFR α -expressing preadipocytes. We described these in the revised manuscript (p. 12).

3. Fig 6 only shows that Rnf20 OE upregulates WAT markers (6f-g). Could the authors also show the gene/protein expression of thermogenic markers/fatty acid oxidation in the whole iWAT (a couple of transcription factors and downstream targets)?

Thanks for the comment. We examined the expression levels of thermogenic genes and fatty acid

oxidation genes in RNF20 OE in iWAT. Our data showed that mRNA levels of thermogenic genes (**Fig. 5g and new Fig. 5h**) and fatty acid oxidation genes (**new Extended Data Fig. 7c**) were upregulated by RNF20 OE in iWAT. Conversely, iWAT-specific RNF20 KD reduced mRNA levels of thermogenic genes and fatty acid oxidation genes (**new Fig. 5i–k and Extended Data Fig. 8c**). We described these in the revised manuscript (pp. 10–11).

4. Likewise, figs 7i-k could also benefit from the inclusion of thermogenic markers.

Following the reviewer's suggestion, we examined the level of UCP1 protein. The level of RNF20 protein in AdipoChaser mice was upregulated by RNF20 OE (**new Extended Data Fig. 10j**). We described these in the revised manuscript (pp. 12–13).

5. Did the authors measure the impact of iWAT Rnf20 OE on whole-body insulin sensitivity or glucose tolerance?

We appreciate this comment. We examined wholebody insulin sensitivity (i.e. insulin tolerance test, ITT) and glucose tolerance (i.e. glucose tolerance test, GTT) in iWAT RNF20 OE mice before and after cold exposure (experimental scheme in **new Extended Data Fig. 7d**). Interestingly, we found that iWAT RNF20 OE appeared to enhance wholebody insulin sensitivity, especially in the late phase of ITT (60 m after an insulin injection), whereas iWAT RNF20 OE did not largely affect wholebody glucose tolerance (**new Extended Data Fig. 7e–l**). In addition, RNF20 OE enhanced phosphorylation of GSK3 β in iWAT upon insulin with increased UCP1 (**new Extended Data Fig. 7m**). Nonetheless, additional research is necessary to elucidate the underlying mechanism of these data. We described these in the revised manuscript (p. 10).

6. Are the plasmids specifically designed to target adipocyte progenitors or mature adipocytes at the injection site? Plasmid information is unavailable in the methods.

We added plasmid information in the Methods section (p. 28–29). As we used cytomegalovirus (CMV) promoter-driven expression vectors, our plasmids would not be selectively overexpressed in adipocytes or beige precursor cells in the injected tissue. In the future, we plan to generate cell-type-specific overexpression vector system.

Minor comments

1. Some samples have been labelled as “ct-27, ct29, ct 33, etc.” in some figures, which is confusing.

To avoid confusion, we replaced the position of the Ct value display and added further explanations of the Ct value in the legend.

2. It is hard to understand the blot data (e.g., 2e, 5h, 5i, 1l, 6e, 6f, 7h). The authors could have used a higher % gel to separate the smaller kDa proteins.

According to the reviewer's suggestion, we re-performed Western blotting analyses using higher % SDS-PAGE gels (**new Fig. 11, 2e, 5h, 5m, 6e, and 6f**).

3. Symbols used in the extended data need to be fixed.

We fixed symbols in the **new Extended Data Figure 1b**.

4. Line 198, *Mtif2* assay, is not written in detail in the methods/figure legend/results sections.

We added the information on *Mtif2* luciferase assay in the revised manuscript (p. 24–25).

5. How 72 h cold exposure time point was adopted?

Upon acute cold stimuli (e.g. 6 h), iWAT appeared to activate the thermogenic gene program with elevated *Ucp1* mRNA (**new Extended Data Fig. 7a**). However, the amount of UCP1 protein in iWAT was not significantly increased at the acute phase of cold stimuli (6 h cold) (**new Fig. 5b and Reviewer's only Fig. 2**). Also, cold stimuli for 24 h did not significantly increase the level of UCP1 protein in iWAT (**Reviewer's only Fig. 2**). Upon cold stimuli for 72 h, the level of UCP1 protein in iWAT was consistently and significantly elevated, and multilocular beige adipocytes were manifested, which are consistent with a previous report that *de novo* beige adipogenesis is active in 72 h cold¹⁰. Based on these data, we adopted 72 h cold exposure for prolonged cold stimuli to examine beige adipogenesis in iWAT. We modified the revised manuscript (p. 10).

Reviewer's only Fig. 2. The protein level of UCP1 protein level in iWAT during cold stimuli (6°C).

Reviewer #3 (Remarks to the Author):

The authors showed that RNF20 decreases during acute cold and enhances thermogenic activity in BAT, while RNF20 increases during chronic cold in iWAT. Furthermore, the authors showed that the regulation of thermogenesis in BAT is mediated by the accumulation of GABP α .

The RNF20-GABP α axis in BAT thermogenesis is a very important finding. However, the molecular mechanisms mediated by GABP α and NCoR1 and the function of RNF20 in iWAT are not sufficiently substantiated, as most of them are based on overexpression systems. Numerous additional key direct data are needed to support the main claims of the paper.

Major concern:

1. Adipocyte-specific KO mice should be used for analysis. Most of the data is due to OE, especially in experiments using iWAT, and at least KO and siRNA should be used in the analysis.

Fig.5: Please verify whether thermogenic activity is enhanced during Chronic cold in iWAT.

Fig.6: How about in RNF20KO/KD and NCoR1KO/KD conditions?

Fig.7i-k: How about in RNF20KO/KD conditions?

Thanks for the critique for animal models. As we mentioned above [the reply for the comment No.1 from Reviewer 2], we fully recognize the significance of adipose-specific *Rnf20* KO mouse model to enhance the value of our findings. Despite our best efforts over several years, we have encountered technical hurdles in generating adipocyte-specific, brown adipocyte-specific, or preadipocyte-specific *Rnf20* deletion mouse models. While we were preparing this manuscript, other group has recently reported that RNF20-NCoR1 axis regulates white adipogenesis with their adipocyte-specific *Rnf20* deletion model¹¹, consistent with our previous report¹². As an alternative approach, we have decided to modulate *Rnf20* expression specifically in BAT or iWAT through direct nucleotide injection and utilized cell line models to study mechanistic understanding. These approaches made us and others to investigate cell- or tissue-specific gene functions^{3,7-9}. Without any doubt, we strongly agree with the reviewer's comment that future studies are required for generating and employing brown adipocyte-specific or beige precursor-specific *Rnf20* gain- and loss-of-function mouse models. We added this limitation in the revised manuscript (p. 16).

First (related to Fig. 5), to verify thermogenic activity was enhanced in iWAT upon chronic cold stimuli, we examined the mRNA and protein levels of UCP1, the crucial mediator for thermogenesis^{13,14}. Upon chronic cold stimuli, the mRNA and protein levels of UCP1 were upregulated (**new Fig. 5b and Extended Data Fig. 7a**). In particular, the mRNA level of *Ucp1* in iWAT upon chronic cold was comparable to the mRNA level of *Ucp1* in BAT (Ct value 20 in iWAT *Ucp1* upon chronic cold and Ct value 21 in BAT *Ucp1* in RT. **new Extended Data Fig. 7a and Fig. 2d**). These data indicated that thermogenesis might be enhanced in iWAT upon chronic cold stimuli. We described these in the revised manuscript (p. 10).

Second (related to Figs. 5 and 6), we conducted iWAT-specific *Rnf20* KD experiments by delivering siRNA against *Rnf20* in a contralateral manner (**new Fig. 5i**). As shown in **new Fig. 5j, k, and Extended Data Fig. 8a–c**, iWAT-specific *Rnf20* KD downregulated iWAT temperature, the number of multilocular beige adipocytes, and the mRNA levels of thermogenic and fatty acid oxidation genes. In

line herewith, iWAT of *Rnf20* heterozygous-null mice upon chronic cold stimuli showed attenuated thermogenic gene expression (**Extended Data Fig. 8d–f**). We described these in the revised manuscript (pp. 10–11). Further, we performed iWAT-specific *Ncor1* KD experiments via siRNA in a contralateral manner. iWAT-specific *Ncor1* KD slightly but substantially elevated iWAT temperature compared to control upon chronic cold stimuli (**new Extended Data Fig. 9d**). Moreover, *Rnf20* and *Ncor1* double KD in iWAT partly nullified the enhanced thermogenic effect of *Ncor1* KD on iWAT thermogenesis (**new Extended Data Fig. 9k**), indicating that RNF20-NCoR1 axis in iWAT would regulate thermogenic activity upon chronic cold stimuli. We described these in the revised manuscript (pp. 11–12).

Lastly (related to Fig. 7i–k), we suppressed RNF20 expression in AdipoChaser mouse model via siRNA in a contralateral manner. In iWAT-selective *Rnf20* KD mice upon chronic cold, *de novo* beige adipogenesis and the level of UCP1 protein were decreased (**new Extended Data Fig. 10k, l**). We described these in the revised manuscript (p. 13).

2. Fig.1a, 1b, 4n, 5b, 6b, 7b Extended Data Fig.2a: The analysis should be performed with $N \geq 3$.

Following the reviewer's suggestion, we performed Western blotting analyses with $N \geq 3$ (**new Fig. 1a, 1b, 4n, 4o, 5b, 6b, and 7b**). For Extended Data Fig. 2a (in the revised version, **Extended Data Fig. 2b**), we deleted the quantitative graph.

3. Fig.2g, 2i, 2o, 2q, 3o: The BAT Mock and BAT siNC data should be equivalent if they are measured under the same conditions. If they are analyzed under different conditions, they should be clearly indicated, and if not, they should be reanalyzed under more rigorous conditions or with a greater number of analyses.

Thanks for this critique. In the previously submitted manuscript, each experiment was performed on different days. Thus, it is likely that the difference in the experimental environments led to the different outcomes of the control groups. To overcome this issue, we newly conducted BAT Mock, RNF20 OE, and GABP α OE experiments on the same day (**new Fig. 2g, 2o, 3o**), and BAT siNC and si*Rnf20* experiments (**new Fig. 2i, 2q**) on the same day. In the new experiments, the rectal temperatures of the control groups were similar. Further, consistent with the data of the previously submitted manuscript, in BAT, RNF20 OE downregulated rectal temperature whereas RNF20 KD upregulated rectal temperature upon cold exposure (**new Fig. 2g, 2i, 2o, 2q, 3o**). We described these in the revised manuscript (p. 19).

4. It is not certain whether GABP α undergoes ubiquitin-dependent degradation in BAT. The band of ubiquitination in Fig. 4k does not appear to be the typical smear/ladder-like band. It is not possible to evaluate this ubiquitination assay because the experimental conditions are not described, but when detecting by ubiquitin antibody, proteins that co-immunoprecipitate must be eliminated by denaturing conditions or other means to be evaluated correctly. That is, it may be overestimated due to contamination of ubiquitination of the binding proteins. One method is to perform a pull-down with TUBE (Tandem ubiquitin binding entity) and evaluate it by substrate (i.e. GABP α) antibody.

When performing under RNF20OE conditions, please add RNF20 deltaRING mutant, and also perform under KO/KD conditions.

Thanks for the suggestion. To follow the comment, we tried to utilize TUBE (tandem ubiquitin binding entity) assay. Unfortunately, TUBE was no longer supplied by the manufacturers (<https://www.rndsystems.com/products/recombinant-human-ubiquitin-1-tandem-uba-tube2-agarose-cf-am-130>). Very recently, we have noticed the existence of TUBE-2 in¹⁵. However, we ran out of time to conduct the experiments with TUBE-2. As an alternative, we have performed a ubiquitination assay using an antibody against mono- and polyubiquitinated conjugates, which is widely used to detect polyubiquitin^{16,17} (mono- and polyubiquitinated conjugates; ENZ-ABS840; Enzo Life Sciences). Also, we increased the number of biological replicates (*N*) in the ubiquitination assay for the reliability of the data. The data from the re-performed experiments were somewhat consistent with those of the initially submitted manuscript. BAT RNF20 OE showed a tendency to increase GABP α polyubiquitination with a smear/ladder-like band pattern (**new Fig. 4k**), while BAT RNF20 KD seemed to reverse it (**new Fig. 4l**). On the other hand, iWAT RNF20 OE showed no significant effects on the polyubiquitination of GABP α (**new Fig. 5m**). Even though we tried to construct RNF20 deltaRING mutant, we had hard time to construct the deltaRING mutant within a limited duration of revision. We replaced the data and described co-immunoprecipitation methods in the revised manuscript (pp. 21–22)

5. Regarding BAT-specific decrease of GABP α , authors point to differences in expression levels as one possibility. Does overexpression of GABP α in iWAT under chronic cold conditions lead to GABP α degradation?

We appreciate this comment. Previously, we proposed that the low level of GABP α protein in iWAT would be one of the reasons why RNF20 would not significantly promote polyubiquitination and degradation of GABP α protein in iWAT. We believe that the reviewer's suggestion is an excellent method to investigate whether RNF20 might promote GABP α polyubiquitination if GABP α protein is abundant in iWAT. Following the reviewer's comment, we overexpressed GABP α in iWAT and then exposed them to chronic cold. When the level of *Gabpa* mRNA was increased in iWAT GABP α OE (**new Extended Data Fig. 11a**), the degree of polyubiquitination of GABP α in iWAT was clearly enhanced (**new Extended Data Fig. 11b**). These data suggest that the level of GABP α would be one of the determinant factors whether or not GABP α would be one of the preferential substrates for E3 ubiquitin ligases such as RNF20. We described these in the revised manuscript (p. 16).

6. Further investigation is needed to determine whether GABP α undergoes degradation in iWAT or not. There is a possibility that synthesis and degradation are balanced. First, for degradation, please perform the same experiment as Fig. 4n under normal and cold conditions with RNF20+/-BAT and iWAT. Please measure Gabpa mRNA levels in iWAT cells under normal, Chronic cold, and beige adipogenesis conditions with respect to synthesis.

Thanks for the comment. Practically and technically, it is very difficult to measure *in vivo* GABP α protein synthesis and degradation in tissue samples. Instead, we performed a cycloheximide-chase assay in differentiated brown adipocytes under RNF20 OE and KD with three independent experiments. As

shown in **new Fig. 4n and 4o**, RNF20 OE potentiated the rate of GABP α protein degradation, whereas RNF20 KD repressed GABP α protein degradation, implying that RNF20 would promote the protein degradation of GABP α in brown adipocytes.

Next, we performed a cycloheximide-chase assay in mature beige adipocytes derived from PDGFR α -expressing (CD31⁻CD45⁻PDGFR α ⁺) preadipocytes. In mature beige adipocytes, RNF20 OE and KD did not significantly alter GABP α protein stability (**new Extended Data Fig. 8k, l**). Also, in iWAT, when RNF20 OE slightly elevated GABP α mRNA and protein levels, RNF20 OE did not greatly potentiate GABP α polyubiquitination in iWAT (**new Fig. 5l-n**), implying that RNF20 might not stimulate protein degradation of GABP α in iWAT.

According to the reviewer's comment, we examined mRNA and protein levels of GABP α in iWAT under cold challenge as well as beige adipogenic conditions. As shown in **new Fig. 5b and Extended Data Fig. 7a**, both mRNA and protein levels of GABP α appeared to be upregulated by chronic cold exposure (mRNA: 1.7-fold, protein: 1.5-fold). During beige adipogenesis, both mRNA and protein levels of GABP α were upregulated (**new Extended Data Fig. 10a, b**, mRNA: 1.3-fold, protein: 1.8-fold). We described these in the revised manuscript (pp. 9–12).

7. Ubiquitin-dependent degradation of NCoR1 has been partially implicated in HEK293T and 3T3-L1 cells in a previous report by the authors (ref. 25). In line with the author's claim that RNF20 ubiquitin-dependent degradation of GABP α only in specific cell populations, the ubiquitin-dependent degradation of NCoR1 must be proven in each cell type. Since the authors claim that NCoR1 is a substrate of RNF20 in iWAT, please prove ubiquitination-dependent degradation, at least in this iWAT. In addition, how do NCoR1 protein and mRNA levels change in iWAT RNF20KO/KD under normal, chronic cold, and beige adipogenesis conditions?

Line324-326: I may have missed it, but I could not identify any data where NCoR1 undergoes ubiquitination-dependent degradation in BAT, at least in refs. 25 and 26. Is there any other evidence? How do NCoR1 protein and mRNA levels change under normal and acute cold conditions in BAT? And does it vary by RNF20KO/KD?

Thanks for these comments. To investigate whether the level of RNF20 would mediate polyubiquitination of NCoR1, we pulled down NCoR1 and examined the levels of ubiquitination in iWAT Mock or iWAT RNF20 OE. As shown in **new Extended Data Fig. 9e**, the band intensity of polyubiquitin of NCoR1 was enhanced by RNF20 OE. In addition, we examined mRNA and protein levels of NCoR1 in iWAT RNF20 OE and KD. While mRNA levels of NCoR1 were not significantly different between iWAT Mock and iWAT RNF20 OE, the protein level of NCoR1 was evidently reduced (**Fig. 6f, g**). In line with these, whereas mRNA levels of *Ncor1* were not altered in iWAT of RNF20 suppression mouse models such as iWAT RNF20 KD and *Rnf20* heterozygous-null mice, NCoR1 protein levels were greatly upregulated (**new Extended Fig. 9f-h**), implying that RNF20 would potentiate polyubiquitination and degradation of NCoR1 in iWAT.

Following the comment, we investigated mRNA and protein levels of NCoR1 during cold stimuli and beige adipogenesis. Whereas mRNA levels of *Ncor1* in both conditions were not largely changed (**new Extended Fig. 7a and 10a**), the protein levels of NCoR1 were downregulated upon cold stimuli and

beige adipogenic stimuli, which were inversely correlated with the level of RNF20 protein (**new Fig. 6b and Extended Data Fig. 10b**). Together, these data propose that RNF20 would regulate NCoR1 protein levels in iWAT.

Next, we examined the role of RNF20 on NCoR1 protein abundance in BAT. While mRNA levels of *Ncor1* in BAT were not different in RNF20 gain- and loss-of-function in BAT compared to controls, the protein levels of NCoR1 were decreased by RNF20 OE, and conversely, increased by RNF20 KD (**new Extended Data Fig. 5j–o**). Further, upon cold stimuli, BAT NCoR1 protein levels, but not mRNA levels, were upregulated, which was negatively correlated with RNF20 protein levels (**new Fig. 1b and Extended Data Fig. 2a, b**). These data suggest that RNF20 would regulate NCoR1 protein level in BAT. We described these in the revised manuscript (pp. 5, 10–12).

8. Fig.1k, 1l: Ucp1 mRNA is increased about 1.3-fold by BAT RNF20 +/-, but UCP1 protein appears to be increased more than 2-fold. Fig.2d, 2e: Ucp1 mRNA is reduced about half by BAT RNF20 OE, but UCP1 protein is almost lost. Ucp1 seems to be under control in the process of translation and beyond, how would the author explain it?

We appreciate this comment. Since we used 10% acrylamide SDS-PAGE gel to detect UCP1 protein in the previously submitted manuscript, it is likely that this result would be due to low resolution with a low gel percentage. To resolve this, we conducted Western blotting analyses using a 12% gel. The new data showed that the level of UCP1 protein was similarly increased to the level of mRNA in *Rnf20* heterozygous-null BAT and RNF20 KD BAT (**new Fig. 1l and 2e**). We replaced the figures in the revised manuscript.

Minor comments:

Fig.2: Authors should perform qPCR of Ucp1 and Pgc1a under siRNF20 or RNF20+/- conditions.

Thanks for the suggestion. We added qRT-PCR data of *Rnf20*, *Ucp1*, and *Pgc1a* in BAT siNC and BAT si*Rnf20* (**new Extended Data Fig. 4h**).

Fig.3p: Authors should perform qPCR of Ucp1 and Gabpa under siRNF20 or RNF20+/- conditions.

We added qRT-PCR data of *Ucp1* and *Gabpa* in BAT siNC and BAT si*Rnf20* from WT and *Rnf20* heterozygous-null mice (**new Extended Data Fig. 6g**).

Fig.5b, 6b: Authors should perform qPCR of RNF20, Ucp1, Gabpa, and NCoR1 under iWAT/chronic cold condition.

We added qRT-PCR data of *Rnf20*, *Ucp1*, *Gabpa*, and *Ncor1* in iWAT under chronic cold stimuli (**new Extended Data Fig. 7a**).

Fig.7b: Authors should perform qPCR of RNF20, Ucp1, Gabpa, and PLIN1 under iWAT/beige adipogenesis condition.

We added qRT-PCR data of *Rnf20*, *Ucp1*, *Gabpa*, *Ncor1*, and *Plin1* in beige preadipocytes during beige adipogenesis (**new Extended Data Fig. 10a**).

Fig.4j, 4k: Please add IB: GABPA

We added immunoblotting data of GABP α (new Fig. 4j–l).

Fig.7h: Please add IB: RNF20

In the revised manuscript, we replaced SVF-derived beige adipocytes with PDGFR α ⁺ preadipocyte-derived beige adipocytes. When we harvested PDGFR α ⁺ preadipocytes from WT and *Rnf20* heterozygous-null mice, the cell numbers were not sufficient to carry out Western blotting analysis. For consistency, we deleted SVF-driven beige adipocyte data in the revised manuscript.

References

- 1 Sarvari, A. K. *et al.* Plasticity of Epididymal Adipose Tissue in Response to Diet-Induced Obesity at Single-Nucleus Resolution. *Cell Metab* **33**, 437-453 e435 (2021). <https://doi.org:10.1016/j.cmet.2020.12.004>
- 2 Emont, M. P. *et al.* A single-cell atlas of human and mouse white adipose tissue. *Nature* **603**, 926-933 (2022). <https://doi.org:10.1038/s41586-022-04518-2>
- 3 Nahmgoong, H. *et al.* Distinct properties of adipose stem cell subpopulations determine fat depot-specific characteristics. *Cell Metab* **34**, 458-472 e456 (2022). <https://doi.org:10.1016/j.cmet.2021.11.014>
- 4 Hepler, C. *et al.* Identification of functionally distinct fibro-inflammatory and adipogenic stromal subpopulations in visceral adipose tissue of adult mice. *eLife* **7**, e39636 (2018). <https://doi.org:10.7554/eLife.39636>
- 5 Merrick, D. *et al.* Identification of a mesenchymal progenitor cell hierarchy in adipose tissue. *Science* **364**, eaav2501 (2019). <https://doi.org:10.1126/science.aav2501>
- 6 Han, X. *et al.* A suite of new Dre recombinase drivers markedly expands the ability to perform intersectional genetic targeting. *Cell Stem Cell* **28**, 1160-1176 e1167 (2021). <https://doi.org:10.1016/j.stem.2021.01.007>
- 7 Nunes, A. D. C. *et al.* miR-146a-5p modulates cellular senescence and apoptosis in visceral adipose tissue of long-lived Ames dwarf mice and in cultured pre-adipocytes. *Geroscience* **44**, 503-518 (2022). <https://doi.org:10.1007/s11357-021-00490-3>
- 8 Lee, G. *et al.* SREBP1c-PARP1 axis tunes anti-senescence activity of adipocytes and ameliorates metabolic imbalance in obesity. *Cell Metab* (2022). <https://doi.org:10.1016/j.cmet.2022.03.010>
- 9 Xie, J. *et al.* Akkermansia muciniphila protects mice against an emerging tick-borne viral pathogen. *Nat Microbiol* **8**, 91-106 (2023). <https://doi.org:10.1038/s41564-022-01279-6>
- 10 Wang, Q. A., Tao, C., Gupta, R. K. & Scherer, P. E. Tracking adipogenesis during white adipose tissue development, expansion and regeneration. *Nat Med* **19**, 1338-1344 (2013). <https://doi.org:10.1038/nm.3324>
- 11 Liang, X. *et al.* Rnf20 deficiency in adipocyte impairs adipose tissue development and thermogenesis. *Protein Cell* **12**, 475-492 (2021). <https://doi.org:10.1007/s13238-020-00770-2>
- 12 Jeon, Y. G. *et al.* RNF20 Functions as a Transcriptional Coactivator for PPARgamma by Promoting NCoR1 Degradation in Adipocytes. *Diabetes* **69**, 20-34 (2020). <https://doi.org:10.2337/db19-0508>
- 13 Cannon, B. & Nedergaard, J. Brown adipose tissue: function and physiological significance. *Physiol Rev* **84**, 277-359 (2004). <https://doi.org:10.1152/physrev.00015.2003>
- 14 Nedergaard, J. & Cannon, B. UCP1 mRNA does not produce heat. *Biochimica et biophysica acta* **1831**, 943-949 (2013). <https://doi.org:10.1016/j.bbalip.2013.01.009>
- 15 González, A. *et al.* Ubiquitination regulates ER-phagy and remodelling of endoplasmic reticulum. *Nature* **618**, 394-401 (2023). <https://doi.org:10.1038/s41586-023-06089-2>
- 16 Moore, T. M. *et al.* Parkin regulates adiposity by coordinating mitophagy with mitochondrial biogenesis in white adipocytes. *Nat Commun* **13**, 6661 (2022). <https://doi.org:10.1038/s41467-022-34468-2>
- 17 Szymanska, K. *et al.* Regulation of canonical Wnt signalling by the ciliopathy protein MKS1 and the E2 ubiquitin-conjugating enzyme UBE2E1. *eLife* **11**, e57593 (2022). <https://doi.org:10.7554/eLife.57593>

REVIEWER COMMENTS

Reviewer #1 (Remarks to the Author):

The authors have addressed my remarks in a thorough and satisfactory fashion. I have no further comments.

Reviewer #2 (Remarks to the Author):

I appreciate the authors' efforts to address my comments, including isolating specific adipocyte progenitor subsets and tolerance tests. However, I have a few more questions.

1. It is customary to incubate the adipocytes in the maintenance media containing only insulin and T3 post-adipogenic induction. Could the authors provide a reason for including the PPAR γ agonist, rosiglitazone, in the maintenance media? The in-vitro experiments showing Ncor1 degradation due to RNF20 overexpression and all thermogenesis-related experiments must be performed without PPAR γ agonist.
2. In Figure 2, after how many days post Rnf20 OE plasmid or siRNA delivery in vivo the experiments were performed? Also, provide such information for figure 3n-o.
3. Ncor1 blots are unclear in interpreting the data. The authors could have used a lower percentage gel. Pull-down experiments would instead be performed to show the degradation of Ncor1 post-Rnf20 OE plasmid administration in vivo or in vitro in cultured iWAT adipocytes.
4. Most blots are unclear to interpret. The authors could have used low and high-exposure settings. For instance, the authors have performed densitometry in Figures 1b, 2c, 5m, 6j, etc. I wonder how it could be possible to measure the intensity when the bands are saturated and overlapping with the adjacent lanes.
5. Line 40 – rewrite for clarity.
6. It is hard to interpret the Ucp1 blot in the rebuttal doc as the centre area of the blot is brighter than the sides.
7. Did the authors use whole adipose tissue while performing gene and protein expression measurements ex vivo?

Reviewer #3 (Remarks to the Author):

The authors responded well to my questions. Only two remaining point:

Response to major comment 7: Based on additional experiments performed by the authors, it appears that NCoR1 is a substrate for RNF20 in both BAT and iWAT and this relationship does not appear to be much cell type-specific. The only cell type-specific substrate obtained in this study is GABPa in BAT, and the description (Page 2, line 33-35 and Fig. 8) that there are primary substrates in both BAT and iWAT would be misleading. It would be good to describe these things in the discussion appropriately.

Response to minor comment 3: It is very interesting that RNF20 is differentially regulated by cold stimuli in each of BAT and iWAT. Considering the results of extra experiments performed by the authors and Extended Data Fig. 3d, it seems that both are regulated in opposite ways to each other at the protein level rather than at the transcriptional level. Although it is beyond the scope of this study to elucidate the mechanism, it would be good to discuss it in more detail (page 17, line 352-354). For example, in Fig. 8, RNF20 is illustrated as being inhibited by PKA, but was the basis for this stated anywhere?

REVIEWER COMMENTS

Reviewer #1 (Remarks to the Author):

The authors have addressed my remarks in a thorough and satisfactory fashion. I have no further comments.

We are pleased to hear that you have no further comments.

Reviewer #2 (Remarks to the Author):

I appreciate the authors' efforts to address my comments, including isolating specific adipocyte progenitor subsets and tolerance tests. However, I have a few more questions.

1. It is customary to incubate the adipocytes in the maintenance media containing only insulin and T3 post-adipogenic induction. Could the authors provide a reason for including the PPAR γ agonist, rosiglitazone, in the maintenance media? The in-vitro experiments showing Ncor1 degradation due to RNF20 overexpression and all thermogenesis-related experiments must be performed without PPAR γ agonist.

Given that beige adipocyte differentiation is stimulated by PPAR γ activation, we decided to include rosiglitazone in the maintenance media to maximize beige adipogenesis. Although the differentiation protocol adding rosiglitazone in maintenance media has been used by several groups such as Bruce Spiegelman¹, Patrick Seale², Camilla Scheele³, and Prashant Rajbhandari⁴, other groups including Shingo Kajimura⁵, Rana Gupta⁶, and Silvia Corvera⁷ have not added rosiglitazone in maintenance media for beige adipogenesis. As this reviewer might be concerned, we cannot rule out the possibility that rosiglitazone would stimulate thermogenic gene expression and affect NCoR1 protein stability. Thus, following this Reviewer's comment, we cultured the cells with only insulin and T3 in the beige maintenance media, and then, we examined the expression levels of thermogenic genes in differentiated beige adipocytes with or without RNF20 overexpression and knockdown. As shown in **new Extended Data Fig. 10f–i**, RNF20 overexpression potentiated thermogenic gene expression, and conversely, RNF20 knockdown downregulated this, which was consistent with our initial data. Further, we examined whether RNF20 stimulated NCoR1 degradation in beige adipocytes differentiated without rosiglitazone in maintenance media (**new Extended Data Fig. 9f**), which was consistent with our initial data. We described these in the revised manuscript (p. 21).

2. In Figure 2, after how many days post Rnf20 OE plasmid or siRNA delivery in vivo the experiments were performed? Also, provide such information for figure 3n-o.

In **Fig. 2**, the *in vivo* experiments were performed on day 3 after RNF20-expressing plasmid or siRNA delivery. Similarly, in **Fig. 3n–o**, the experiments were conducted on day 3 after GABP α -expressing plasmid injection. This information was described in the revised manuscript (pp. 32–33).

3. Ncor1 blots are unclear in interpreting the data. The authors could have used a lower percentage gel. Pull-down experiments would instead be performed to show the degradation of Ncor1 post-Rnf20 OE plasmid administration in vivo or in vitro in cultured iWAT adipocytes.

Thanks for this comment. To get a better resolution of NCoR1 band, we have re-run samples in low percentage (5% acrylamide) of gels (**new Extended Data Fig. 5k and 9j**). Also, we have re-performed the NCoR1 pull-down experiments (**new Extended Data Fig. 9e**). Further, to examine whether RNF20 would stimulate NCoR1 protein degradation, we overexpressed RNF20 in beige adipocytes as described above. As shown in **new Extended Data Fig. 9f**, RNF20 overexpression stimulated protein degradation of NCoR1 in the presence of cycloheximide. We described these in the revised manuscript (p. 12).

4. Most blots are unclear to interpret. The authors could have used low and high-exposure settings. For instance, the authors have performed densitometry in Figures 1b, 2c, 5m, 6j, etc. I wonder how it could be possible to measure the intensity when the bands are saturated and overlapping with the adjacent lanes.

We appreciate this comment. In the previous manuscript, we performed densitometry of the western blots using the “Analyze & Gels” tool of the ImageJ program. As the reviewer pointed out, this approach would lead to compromising the accuracy of quantification of saturated and overlapping bands with the adjacent lanes. Thus, we re-performed Western blot quantification according to the guidelines proposed by Gassmann, et al. ⁸. Briefly, we removed the background using the rolling ball method. Then, we quantified the center 30% of each lane width of the blots to reduce errors caused by overlapping bands with the adjacent lanes. In the revised manuscript, we re-quantified all the Western blot bands following the above method (**new Fig. 1a, 1b, 3p, 4m, 4n, 4o, 5b, 5m, 6b, 6e, 6j, 7b, Extended Data Fig. 2a, 8h, and 10b**). Additionally, we have re-run the western blot of Fig. 2c (**new Fig. 2c**). We described the quantification method in the revised manuscript (p. 23).

5. Line 40 – rewrite for clarity.

Following the reviewer’s comment, we have rephrased line 40 to improve clarity.

6. It is hard to interpret the Ucp1 blot in the rebuttal doc as the centre area of the blot is brighter than the sides.

Reviewer’s only Figure. The protein level of UCP1 protein level in iWAT during cold stimuli (6°C).

Per this comment, we modified the UCP1 blot in the same gel of the rebuttal doc with a lesser bright signal (Reviewer’s only Figure).

7. Did the authors use whole adipose tissue while performing gene and protein expression measurements *ex vivo*?

In this study, whole adipose tissues were subjected to examine gene and protein expression, except when we examined PDGFR α -expressing beige precursors (**Fig. 7 and Extended Data Fig. 8–10**) or floating brown adipocytes (**Extended Data Fig. 2**).

Reviewer #3 (Remarks to the Author):

The authors responded well to my questions. Only two remaining point:

Response to major comment 7: Based on additional experiments performed by the authors, it appears that NCoR1 is a substrate for RNF20 in both BAT and iWAT and this relationship does not appear to be much cell type-specific. The only cell type-specific substrate obtained in this study is GABP α in BAT, and the description (Page 2, line 33-35 and Fig. 8) that there are primary substrates in both BAT and iWAT would be misleading. It would be good to describe these things in the discussion appropriately.

Thanks for the comment. We agree with the point that the cell type-specific substrate of RNF20 would be GABP α in BAT and that NCoR1 would be a substrate for RNF20 in both BAT and iWAT. Nevertheless, in the proposed model (Fig. 8), we would like to emphasize that RNF20 would be downregulated by acute cold, leading to the accumulation and activation of GABP α to facilitate thermogenic activation in BAT. Further, upon chronic cold, RNF20 would be upregulated and promote beige adipogenesis through NCoR1-PPAR γ -axis in iWAT.

In BAT, our data suggest that RNF20 would regulate NCoR1 protein abundance. In BAT, the role of RNF20-NCoR1-PPAR γ axis appears to mainly regulate the expression of genes related to lipid metabolism, such as *Fabp4* and *Cd36*, rather than thermogenesis (Extended Data Fig. 5). Thus, to emphasize the role of RNF20 in thermogenic activation in BAT, we have focused RNF20-GABP α axis in the proposed model. As the reviewer suggested, we modified the discussion (p. 15) and the figure legend (p. 38) in the revised manuscript.

Response to minor comment 3: It is very interesting that RNF20 is differentially regulated by cold stimuli in each of BAT and iWAT. Considering the results of extra experiments performed by the authors and Extended Data Fig. 3d, it seems that both are regulated in opposite ways to each other at the protein level rather than at the transcriptional level. Although it is beyond the scope of this study to elucidate the mechanism, it would be good to discuss it in more detail (page 17, line 352-354). For example, in Fig. 8, RNF20 is illustrated as being inhibited by PKA, but was the basis for this stated anywhere?

We appreciate this comment. In BAT and iWAT, RNF20 was differently regulated by cold stimuli, which would be an interesting topic to study in future. It has been reported that, upon norepinephrine treatment, phosphorylation patterns of many proteins are quite different in brown and beige adipocytes⁹. Further, it has been shown that casein kinase 2 (CK2) would be one of the beige-specific kinases⁹. Interestingly, RNF20 is highly phosphorylated at S136 and S138¹⁰, and *in silico* kinase motif prediction suggests that CK2 might phosphorylate the S136 site of RNF20. In future, it will be necessary to explore

the possibility that several kinase cascades, including CK2, regulate protein abundance and stability of RNF20 in brown and beige adipocytes. We described these in the revised manuscript (p. 17). Also, in this regard, we cannot rule out the possibility that PKA might not primarily mediate RNF20 downregulation in BAT upon cold. Thus, we modified the proposed model in Fig. 8.

References

- 1 Kim, H. *et al.* Irisin Mediates Effects on Bone and Fat via α v Integrin Receptors. *Cell* **175**, 1756-1768 e1717 (2018). <https://doi.org:10.1016/j.cell.2018.10.025>
- 2 Holman, C. D. *et al.* Aging impairs cold-induced beige adipogenesis and adipocyte metabolic reprogramming. *eLife* (2023). <https://doi.org:10.1101/2023.03.20.533514>
- 3 Palani, N. P. *et al.* Adipogenic and SWAT cells separate from a common progenitor in human brown and white adipose depots. *Nat Metab* **5**, 996-1013 (2023). <https://doi.org:10.1038/s42255-023-00820-z>
- 4 Patel, S. *et al.* Mammary duct luminal epithelium controls adipocyte thermogenic programme. *Nature* (2023). <https://doi.org:10.1038/s41586-023-06361-5>
- 5 Oguri, Y. *et al.* CD81 Controls Beige Fat Progenitor Cell Growth and Energy Balance via FAK Signaling. *Cell* **182**, 563-577 e520 (2020). <https://doi.org:10.1016/j.cell.2020.06.021>
- 6 Zhang, Q. *et al.* Distinct functional properties of murine perinatal and adult adipose progenitor subpopulations. *Nat Metab* **4**, 1055-1070 (2022). <https://doi.org:10.1038/s42255-022-00613-w>
- 7 Yang Loureiro, Z. *et al.* Wnt signaling preserves progenitor cell multipotency during adipose tissue development. *Nat Metab* **5**, 1014-1028 (2023). <https://doi.org:10.1038/s42255-023-00813-y>
- 8 Gassmann, M., Grenacher, B., Rohde, B. & Vogel, J. Quantifying Western blots: pitfalls of densitometry. *Electrophoresis* **30**, 1845-1855 (2009). <https://doi.org:10.1002/elps.200800720>
- 9 Shinoda, K. *et al.* Phosphoproteomics Identifies CK2 as a Negative Regulator of Beige Adipocyte Thermogenesis and Energy Expenditure. *Cell Metab* **22**, 997-1008 (2015). <https://doi.org:10.1016/j.cmet.2015.09.029>
- 10 Huttlin, E. L. *et al.* A tissue-specific atlas of mouse protein phosphorylation and expression. *Cell* **143**, 1174-1189 (2010). <https://doi.org:10.1016/j.cell.2010.12.001>

REVIEWERS' COMMENTS

Reviewer #2 (Remarks to the Author):

The authors have addressed my remarks and suggestions. I have no further comments.